# Engineering biosynthesis of the anticancer alkaloid noscapine in yeast

Yanran Li[1] & Christina D. Smolke[1]

Noscapine is a potential anticancer drug isolated from the opium poppy *Papaver somniferum*, and genes encoding enzymes responsible for the synthesis of noscapine have been recently discovered to be clustered on the genome of *P. somniferum*. Here, we reconstitute the noscapine gene cluster in *Saccharomyces cerevisiae* to achieve the microbial production of noscapine and related pathway intermediates, complementing and extending previous *in planta* and *in vitro* investigations. Our work provides structural validation of the secoberberine intermediates and the description of the narcotoline-4′-O-methyltransferase, suggesting this activity is catalysed by a unique heterodimer. We also reconstitute a 14-step biosynthetic pathway of noscapine from the simple alkaloid norlaudanosoline by engineering a yeast strain expressing 16 heterologous plant enzymes, achieving reconstitution of a complex plant pathway in a microbial host. Other engineered yeasts produce previously inaccessible pathway intermediates and a novel derivative, thereby advancing protoberberine and noscapine related drug discovery.

[1] Shriram Center, Department of Bioengineering, Stanford University, 443 Via Ortega, MC 4245, Stanford, California 94305, USA. Correspondence and requests for materials should be addressed to C.D.S. (email: csmolke@stanford.edu).

Plant natural products span a large and diverse class of compounds with critical ecological functions[1] and important therapeutic niches[2,3]. Due to the structural complexity of many of these compounds, chemical synthesis routes have remained challenging, and the supply of plant natural products still largely relies on extraction from plant biomass. Derivatives of target compounds are difficult to obtain from plant-based production platforms because engineering plants to accumulate pathway intermediates and novel functionalized products is challenging[4]. The recent engineering of microbial production platforms for specific plant natural products, such as medicinal opioids, artemisinin and precursors to Taxol and vinblastine[5–11], highlights an alternative approach to sourcing these valuable compounds. However, this approach requires knowledge of the corresponding biosynthetic routes in plants.

Currently, in planta gene silencing studies are used to verify gene identity and identify enzyme function within the pathway; and in vitro biochemical studies are used to understand the function and study enzymatic mechanisms[12,13]. However, for full elucidation of complex pathways these methods can encounter limitations. In planta characterization of individual enzyme activities can be impacted by the complex endogenous protein and metabolite background, and in vitro characterization can be limited by the availability of pathway intermediates as substrates and standards, many of which are difficult to synthesize chemically or extract from producing plants. A number of tools have recently been developed that enable the efficient reconstitution of known plant natural product pathways in the budding yeast Saccharomyces cerevisiae[14–17]. These tools are advancing the reconstitution of complex plant pathways in yeast[18–20], with recent examples comprising up to 13 heterologous enzymatic steps from precursors supplied in yeast[7,8]. Therefore, step-wise reconstitution of plant pathways in yeast – leveraging the integration of recent methodological advances, including inexpensive DNA synthesis, rapid combinatorial pathway assembly strategies[21–23] and biological parts tailored to optimizing expression, folding and processing of plant enzymes[14–17,24] – can be applied to obtain sufficient levels of intermediate metabolites along a pathway for structural verification, and at the same time provide substrate for the subsequent enzymatic steps. This synthetic approach can be used to connect the upstream pathway identification in planta and the downstream in vitro characterization of enzyme mechanism.

Noscapine is a potential anticancer medicine with a long-established safety record as an oral drug[25] that is currently under pre-clinical trials[26–29] and also used off-label as a drug to treat cancer[30]. Some noscapine derivatives such as 9-bromo-noscapine, referred as noscapinoids, exhibit even greater potency than noscapine towards a number of cancer cell lines[31–35]. However, the synthesis of noscapinoids is currently limited to those that can be achieved from noscapine, due to limited access to pathway intermediates such as narcotoline and papaveroxine[31–35]. Noscapine is a phthalideisoquinoline alkaloid synthesized from L-tyrosine through scoulerine[36], and the biosynthesis scheme involves multiple types of alkaloids, including 1-benzylisoquinoline, protoberberine and secoberberine alkaloids[37,38]. Among them, secoberberine alkaloids exhibit a unique structural scaffold, but their pharmacological activities are less investigated than alkaloids such as 1-benzylisoquinoline, protoberberine or phthalideisoquinoline alkaloids[39–46], which may be due to limited access to these molecules.

A 10-gene cluster involved in the biosynthesis of noscapine, 1 was recently discovered in Papaver somniferum[37]. The noscapine gene cluster, which represents one of the most complex plant gene clusters identified to date, was initially characterized by gene silencing experiments in P. somniferum[37] (Fig. 1). Subsequent series of in vitro characterization experiments were able to correct the ordering of several steps (that is, the enzymatic steps catalysed by CYP82X1, CYP82X2 and PsAT1) and proposed an updated biosynthetic pathway[47–50] (Fig. 1); however, the sequence of the final steps in the pathway remained unclear and the enzyme catalysing the conversion of narcotoline to noscapine remained unidentified.

To provide an alternative route to the production of noscapine, and to overcome the inaccessibility of many pathway intermediates and potentially interesting derivatives, we reconstitute the noscapine gene cluster in the yeast host S. cerevisiae. Our synthetic approach provides experimental support for the previously proposed reaction matrix of the final three steps of the pathway and suggests that the remaining 4′-O-methylation step is catalysed by two different O-methyltransferases together, and to the best of our knowledge, is likely to be the first known O-methylation natively catalysed by an O-methyltransferase heterodimer. Our reconstituted noscapine gene cluster also provides sufficient levels of pathway intermediates to support structural verification, which allows us to validate the secoberberine structures synthesized by CYP82X1. In addition, the 14-step biosynthesis of noscapine from the early benzylisoquinoline alkaloid norlaudanosoline is achieved in yeast by optimizing the expression of 16 plant enzymes, reaching a titre of $1.64 \pm 0.38 \mu M$ noscapine.

## Results

**Reconstituting TNMT and CYP82Y1 activities in yeast.** The noscapine gene cluster encodes 10 enzymes to convert (S)-scoulerine to noscapine. The cluster encodes four cytochrome P450 monooxygenases (CYP719A21, CYP82Y1, CYP82X1 and CYP82X2), three methyltransferases (PsMT1, PsMT2 and PsMT3), one carboxylesterase (PsCXE1), one short-chain dehydrogenase/reductase (PsSDR1), and one acetyl transferase (PsAT1), making it one of the most complex plant natural product pathways described to date[37]. The recent gene silencing experiment in P. somniferum[37] and in vitro biochemical characterization studies[50,51] suggest that PsMT1 and CYP719A21 are the scoulerine 9-O-methyltransferase (S9OMT) and canadine synthase (CAS), respectively, catalysing the conversion of (S)-scoulerine to (S)-canadine, 2, which is then converted to N-methylcanadine, 3, by tetrahydroprotoberberine cis-N-methyltransferase from P. somniferum (PsTNMT), and this compound is subsequently hydroxylated to 1-hydroxy-N-methylcanadine, 4, by 1-hydroxy-N-methylcanadine synthase from P. somniferum (CYP82Y1).

We started the reconstitution of the noscapine biosynthetic pathway from canadine, as the biosynthesis of 2, from norlaudanosoline, 12, has previously been engineered in S. cerevisiae[52,53]. The genes related to the synthesis of noscapine from canadine were introduced into S. cerevisiae through either plasmid or yeast artificial chromosome constructs or chromosomal integration, as indicated. The engineered yeast strains were then cultured in synthetic defined medium (SDM) supplemented with 250 μM of racemic 2, which is assumed to contain equal amounts of each enantiomer. The medium is then directly analysed for pathway metabolites after 72 h of growth by liquid chromatography/mass spectrometry (LC/MS) analysis. Previous studies indicated that using galactose as a carbon source improves alkaloid production in yeast from fed substrates, which may be related to the metabolic or transcriptional responses induced by galactose[54]. In addition, since the GAL1 promoter was used in several constructs for reconstituting the biosynthesis of noscapine, galactose was used as the carbon source in all the in vivo assays in this work.

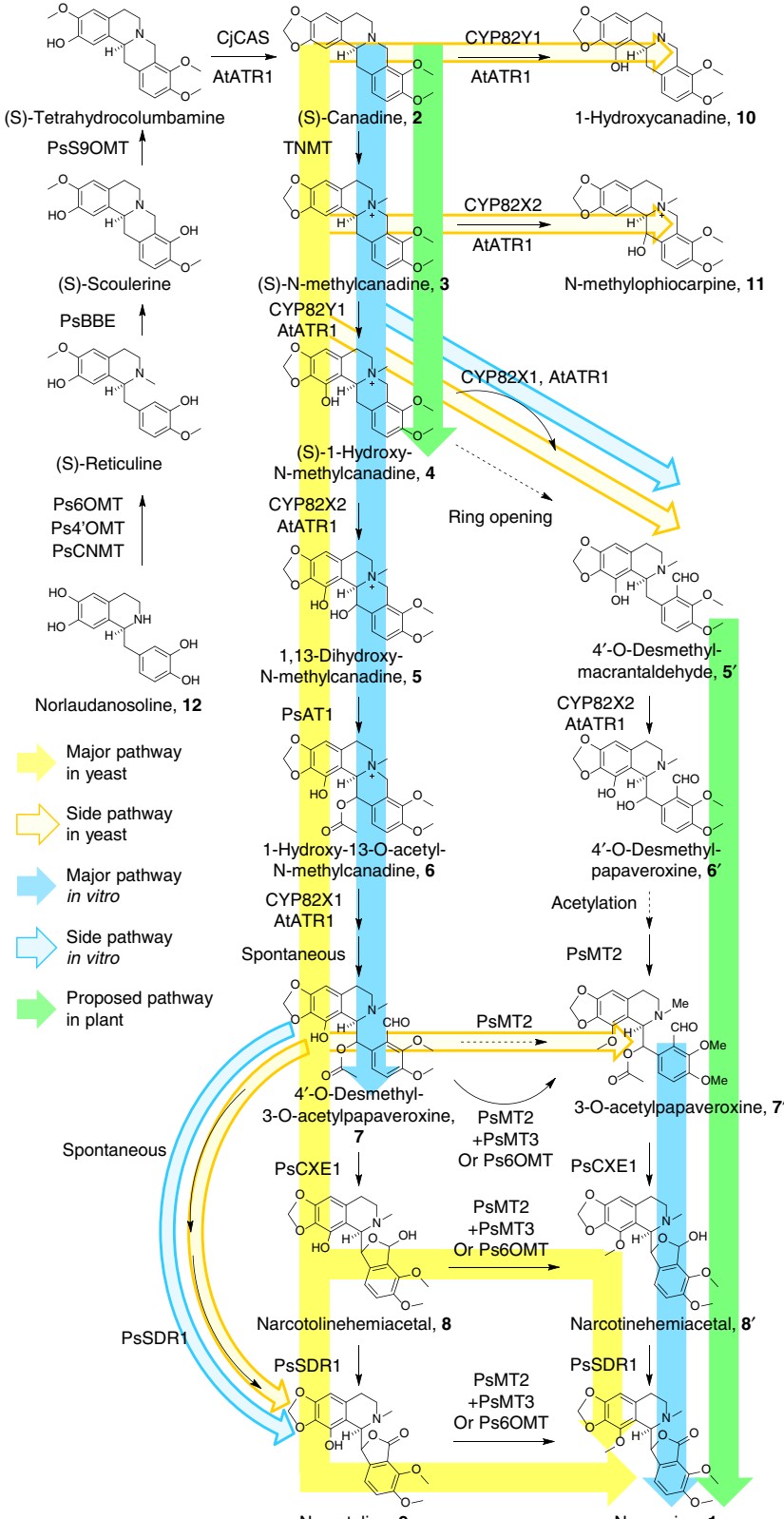

**Figure 1 | Overview of the biosynthetic pathway of noscapine.** Biosynthetic pathways from norlaudanosoline, **12**, to noscapine, **1**, proposed according to *in situ* virus-induced gene silencing experiments in *P. somniferum* (green), *in vitro* characterization (blue) and pathway reconstitution in yeast (yellow). Solid colour: proposed major biosynthetic pathway to noscapine; faded colour: putative side reaction due to enzyme promiscuity; solid arrow: characterized enzymatic step; dashed arrow: unverified enzymatic step.

Yeast strains were engineered to express PsTNMT, CYP82Y1 and a cytochrome P450 reductase partner (CPR) from *Arabidopsis thaliana* (AtATR1) that was previously shown to function with diverse plant P450s (refs 52,55). When grown in the presence of 250 μM fed racemic **2** at 30 °C, the engineered strains synthesized **4** ($m/z^+ = 370$; Figs 1 and 2a and Supplementary Fig. 1c), in agreement with recent *in vitro* biochemical characterizations of TNMT and CYP82Y1 (refs 48,56). Nuclear magnetic resonance (NMR) analyses further confirmed the identity of **4** synthesized in the engineered yeast (Supplementary Fig. 2 and Supplementary Table 1). However, the synthesis of **4** was inefficient (21.9 ± 5.5 μM, Fig. 3a), limited by the CYP82Y1-catalysed step (~20% conversion from ~100 μM **3**) when that enzyme is expressed from the *GAL1* promoter at 30 °C (ref. 57). To ensure that our strains make sufficient levels of **4** for conversion through the remaining downstream pathway steps, we optimized CYP82Y1 activity by testing a combination of different

N-terminal tags, promoters and growth temperatures. While most of the N-terminal tags did not improve CYP82Y1 activity, the N-terminus of a related plant cytochrome P450, (*S*)-*cis*-*N*-methylstylopine-1-hydroxylase (MSH), resulted in slightly enhanced activity from the chimeric protein (26.9 ± 2.0 μM) when expressed downstream of the *GAL1* promoter at 30 °C (Fig. 3a and Supplementary Fig. 3a). Both CYP82Y1 and the MSH N-terminus swapped CYP82Y1 (CYP82Y1A) exhibit higher activity at 25 °C than 30 °C. The enhanced activity in synthesizing **4** at lower temperature ($P = 0.0021$) and the N-terminus engineering ($P = 0.0436$) are both significant through a standard two-way analysis of variance (Fig. 3a, Supplementary Fig. 3a and Supplementary Table 2). The highest CYP82Y1 activity (96.5 ± 3.0 μM) was observed when CYP82Y1A was expressed from either the strong glycolytic *GPD* promoter or the late stage *HXT7* promoter[58] at 25 °C (Fig. 3a and Supplementary Fig. 3a). Our optimization of CYP82Y1 activity allowed us to achieve ~39% conversion from a racemic mixture of **2** to **4**.

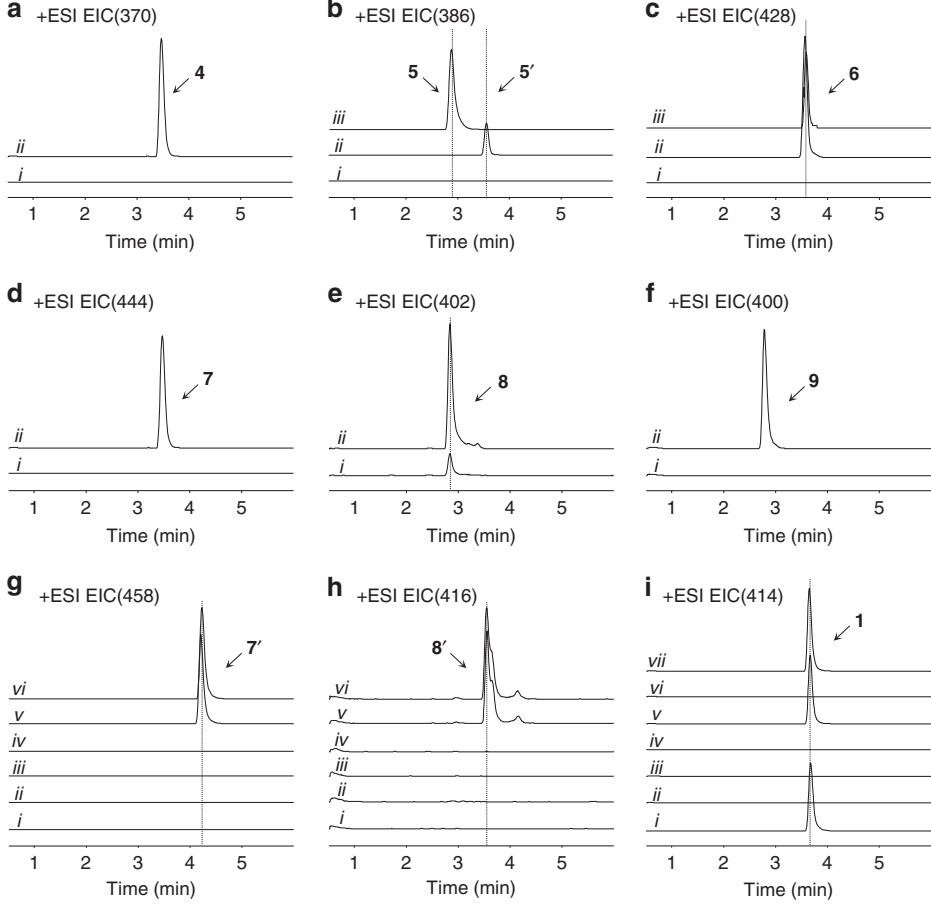

**Figure 2 | LC–MS analysis of yeast strains fed with canadine.** (**a**) Extracted ion chromatogram (EIC) of $m/z^+ = 370$ of (i) **3**-producing yeast strain and (ii) strain expressing PsTNMT, AtATR1 and CYP82Y1. (**b**) EIC of $m/z^+ = 386$ of (i) **4**-producing yeast strain, (ii) strain expressing PsTNMT, AtATR1, CYP82Y1 and CYP82X1 and (iii) strain expressing PsTNMT, AtATR1, CYP82Y1 and CYP82X2. (**c**) EIC of $m/z^+ = 428$ of (i) **5**-producing yeast strain, (ii) strain expressing PsTNMT, AtATR1, CYP82Y1, CYP82X2 and PsAT1 and (iii) **7**-producing strain. (**d**) EIC of $m/z^+ = 444$ of (i) **6**-producing yeast strain and (ii) strain expressing PsTNMT, AtATR1, CYP82Y1, CYP82X1, CYP82X2 and PsAT1. (**e**) EIC of $m/z^+ = 402$ of (i) **7**-producing yeast strain and (ii) strain expressing PsTNMT, AtATR1, CYP82Y1, CYP82X2, PsAT1, CYP82X1 and PsCXE1. (**f**) EIC of $m/z^+ = 400$ of (i) **8**-producing yeast strain and (ii) strain expressing PsTNMT, AtATR1, CYP82Y1, CYP82X2, PsAT1, CYP82X1, PsCXE1 and PsSDR1. (**g**) EIC of $m/z^+ = 458$ of (i) **7**-producing yeast strain, and **7**-producing strain expressing (ii) PsMT2, (iii) PsMT3, (iv) Ps6OMT, (v) PsMT2 and PsMT3, (vi) PsMT2 and Ps6OMT. (**h**) EIC of $m/z^+ = 416$ of (i) **8**-producing yeast strain, and **8**-producing strain expressing (ii) PsMT2 and PsSDR1, (iii) PsMT3, (iv) Ps6OMT, (v) PsMT2, PsMT3 and PsSDR1, (vi) PsMT2, Ps6OMT and SDR1. (**i**) EIC of $m/z^+ = 414$ of (i) noscapine standard, (ii) **9**-producing yeast strain, and **9**-producing strain expressing (iii) PsMT2, (iv) PsMT3, (v) PsMT2 and PsMT3, (vi) PsMT2 and Tf6OMT, (vii) PsMT2 and Ps6OMT. All traces are representative of at least three biological replicates for each engineered yeast strain.

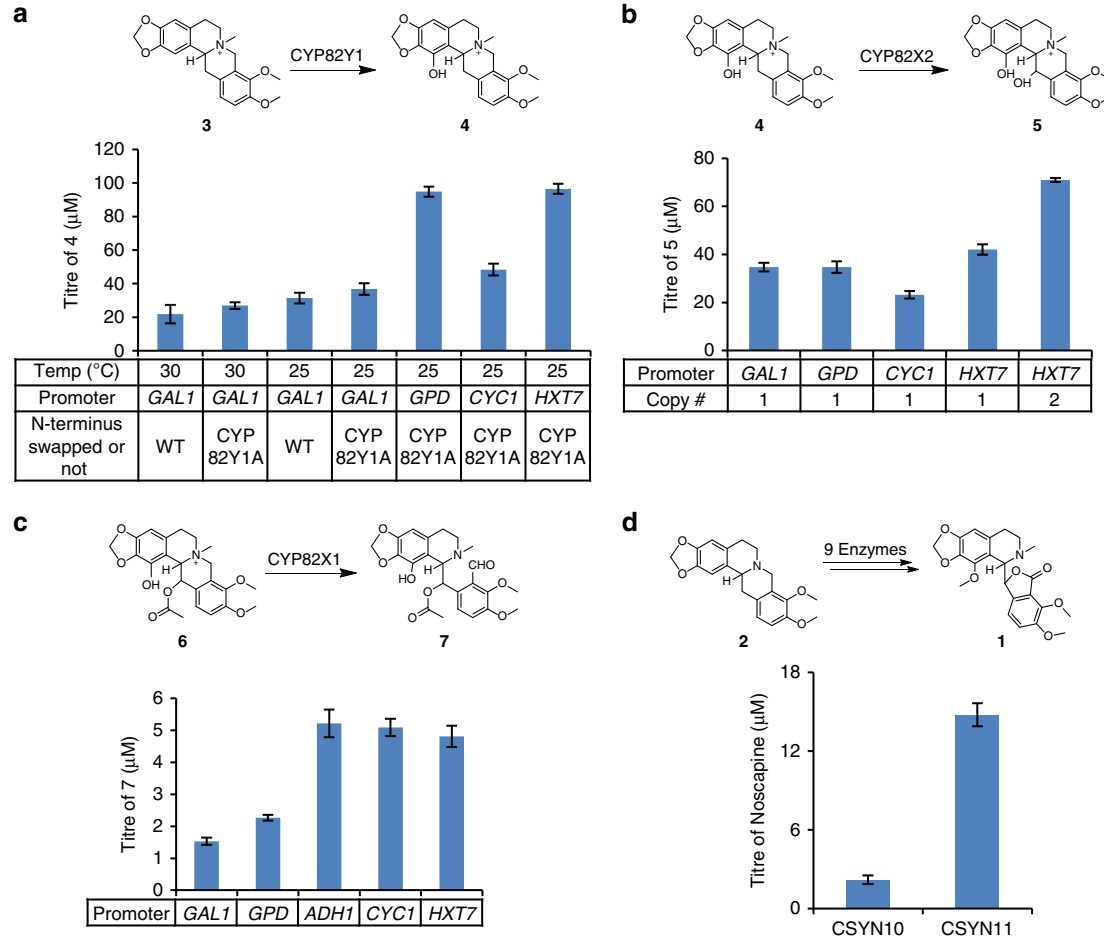

**Figure 3 | Optimization of the activities for three pathway-specific cytochrome P450s.** (**a**) Optimization of CYP82Y1 activity for the synthesis of **4** through varying N-terminal tag, promoter and growth temperature. (**b**) Optimization of CYP82X2 activity for the synthesis of **5** through varying promoter and gene copy number. (**c**) Optimization of CYP82X1 activity for the synthesis of **7** through varying promoter. (**d**) Noscapine titre analysed from engineered strains harbouring different sets of expression cassettes. CSYN10: AtATR1, PsTNMT, CYP82X2 and PsAT1 expressed from the chromosome; CYP82Y1, PsMT2, PsMT3, CYP82X1, PsCXE1 and PsSDR1 expressed from a low-copy plasmid (Supplementary Data 1). CSYN11: AtATR1, PsTNMT, CYP82X2 and PsAT1 expressed from the chromosome; CYP82Y1, PsMT2, PsMT3, CYP82X1, PsCXE1, PsSDR1 and CYP82X2 expressed from a low-copy plasmid (Supplementary Data 1). Bars represent mean values ± 1 s.d. of three biological replicates, and the error bars represent the s.d. of the replicates. CYP82Y1A is the MSH N-terminus swapped CYP82Y1.

**Reconstituting CYP82X2, PsAT1 and CYP82X1 activities.** The pioneering *in planta* studies performed in *P. somniferum* indicated that inactivating CYP82X2 leads to the synthesis of 4′-O-desmethylmacrantaldehyde[37], **5**′. The subsequent *in vitro* study indicated that CYP82X1 is responsible for the conversion of **4** to **5**′ (ref. 47), and the synthesis of **5**′ is due to the substrate promiscuity of CYP82X1. CYP82X2 catalyses the enzymatic step subsequent to CYP82Y1, converting **4** to 1,13-dihydroxy-*N*-methylcanadine, **5**. PsAT1 then acetylates the 13-hydroxy of **5** and affords the synthesis of 1-hydroxy-13-O-acetyl-*N*-methylcanadine, **6**; while CYP82X1 catalyses the structural conversion from protoberberine scaffold of **6** to the secoberberine scaffold of 4′-O-desmethyl-3-O-acetylpapaveroxine, **7** (ref. 47). Secoberberines such as **5**′ and **7** are structurally unique and are currently commercially inaccessible. Thus, few studies have been conducted on secoberberine-based drug development[39–46]. In addition, the structure of **7** synthesized by CYP82X1 has not been confirmed due to the lack of authentic standards and limited quantity of the compound achieved in the *in vitro* investigations[47]. We utilized our optimized **4**-producing strain to achieve the microbial production of secoberberines to resolve the sourcing of this group of compounds.

We first engineered the **4**-producing strain to express CYP82X1 for the synthesis of **5**′. When grown in the presence of **2** at 25 °C, the engineered strain synthesized **5**′ ($m/z^{+} = 386$), which is confirmed by NMR analyses (Fig. 2b, Supplementary Figs 1e and 4 and Supplementary Table 3); however, the efficiency was low ($14.2 ± 0.2\,\mu M$, Supplementary Fig. 3b). We optimized CYP82X1 by testing its activity under different promoters, and observed the highest activity ($22.9 ± 1.5\,\mu M$, Supplementary Fig. 3b) when the enzyme was expressed from a strong constitutive promoter (*PGK1*) at 25 °C (Supplementary Fig. 3b), compared with expression from the *GAL1* promoter (paired *t*-test, $P = 0.00068$).

When CYP82X2 is introduced into the **4**-producing yeast strain, **5** ($m/z^{+} = 386$) (Fig. 2b and Supplementary Fig. 1d) is synthesized, and the identity of **5** was confirmed through NMR analyses (Supplementary Fig. 5 and Supplementary Table 4). We optimized CYP82X2 for the synthesis of **5** by testing its activity under different promoters, and observed the highest activity ($42.1 ± 2.2\,\mu M$) when the enzyme was expressed from the *HXT7* promoter at 25 °C (Fig. 3b and Supplementary Fig. 3c). By engineering more copies of the CYP82X2 expression construct into our **5**-producing strain, the engineered strain exhibits

significantly higher activity towards the synthesis of **5** ($71.0 \pm 0.8\,\mu M$, paired $t$-test, $P = 0.000027$; Fig. 3b). The subsequent addition of PsAT1 in the **5**-producing yeast strain results in the synthesis of **6** ($m/z^{+} = 428$; Fig. 2c and Supplementary Fig. 1g). The identity of **6** synthesized from the engineered yeast strain was confirmed through NMR analyses (Supplementary Fig. 6 and Supplementary Table 5). The expression of PsAT1 from the *GPD* promoter in the optimal **5**-producing strain (Fig. 3b, *HXT7-2* and Supplementary Data 1) leads to the production of **6** at $40.0 \pm 0.5\,\mu M$. The expression of CYP82X1 in the **6**-producing yeast strain leads to the synthesis of **7** ($m/z^{+} = 444$) (Fig. 2d and Supplementary Fig. 1h). Subsequent NMR analyses explicitly indicate the structure of **7** to be 4′-O-desmethyl-3-O-acetylpapaveroxine (Supplementary Fig. 7 and Supplementary Table 6). To ensure that our strains make sufficient **7** for conversion through the remaining pathway steps, we optimized the CYP82X1 for synthesis of **7** by examining its activity under different promoters. The highest activity ($5.2 \pm 0.4\,\mu M$ in the strain harbouring only one copy of CYP82X2 regulated by *HXT7* promoter) was observed when the enzyme was expressed from relatively weak promoters (*ADH1* and *CYC1*) and late promoter (*HXT7*) at $25\,^{\circ}C$ (Fig. 3c and Supplementary Fig. 3d). When we introduced the expression of CYP82X1 regulated by the *HXT7* promoter in the optimal **5**-producing strain (Fig. 3b, *HXT7-2* and Supplementary Data 1), $24.8 \pm 0.5\,\mu M$ of **7** is synthesized from $250\,\mu M$ fed **2**. We observed trace amounts of a compound with $m/z^{+} = 402$ at a retention time (RT) of 2.8 min that agrees with 4′-O-desmethl-papveroxine, **6′**, or narcotolinehemiacetal, **8**, in the **7**-producing strain (Supplementary Fig. 8a,c). This peak was also observed in the **5**-producing strain expressing CYP82X1 (Supplementary Fig. 8a,c). This data implies that **7** may cyclize spontaneously to form **8**, and that CYP82X1 functions on **5** to afford the synthesis of **6′** or cyclized **8**.

The absence of 1,8-dihydroxy-13-O-acetyl-*N*-methylcanadine implies that the conversion from 1,8-dihydroxy-13-O-acetyl-*N*-methylcanadine to **7** is either highly efficient in yeast, or more likely, that this reaction is facilitated by CYP82X1 (Supplementary Fig. 8f). In addition, the acetylation is highly specific towards **5**, and **6** is the only detectable acetylated product in the **7**-producing strain (Fig. 2c). With structures of the products of CYP82X1, **5′** and **7**, explicitly verified by NMR in our study, our results support the previous *in vitro* biochemical characterization of the synthesis of secoberberine that CYP82X2 catalyses the enzymatic step subsequent to CYP82Y1, followed by PsAT1, which is required for CYP82X1 to function efficiently towards the synthesis of noscapine[47]. We hypothesize that the acetylation occurring before the C–N bond cleavage may enable catalysis via a tetrahedral acyl intermediate rather than simple proton abstraction, resulting in more efficient ring formation subsequent to the synthesis of the unstable aldehyde (Supplementary Fig. 8f).

**Reconstituting PsCXE1 and PsSDR1 activities.** According to previous *in planta* gene silencing studies in *P. somniferum*[37], the accumulation of the 4′-O-desmethyl version of the final product noscapine, narcotoline, **9**, is observed when inactivating PsMT2. In addition, inactivating the carboxylesterase and noscapine synthase encoded in the noscapine gene cluster (PsCXE1 and PsSDR1, respectively) resulted in accumulation of 4′-O-methylated 3-O-acetylpapaveroxine, **7′**, and narcotinehemiacetal, **8′**, respectively. However, an *in vitro* study of PsMT2 indicated that PsMT2 has no activity towards narcotoline, but functions as a S9OMT with strict substrate specificity[50]. The recent *in vitro* biochemical study supports the activity of PsCXE1 in the

conversion of **7′** to **8′** and PsSDR1 in the conversion of the $m/z^{+} = 402$ peak synthesized spontaneously from **7** to narcotoline, **9** (ref. 47). Thus, the role of PsMT2 in the pathway (that is, whether it catalyses the narcotoline-4′-O-methylation or (S)-scoulerine-9-O-methylation), the identity of the 4′-O-methyltransferase required for the 4′-O-methylation and the sequence of the enzymatic steps catalysed by the 4′-O-methyltransferase, PsCXE1 and PsSDR1 remain in question.

We first engineered the **7**-producing strain to express PsMT2. However, when grown in the presence of **2** at $25\,^{\circ}C$, the engineered strain did not synthesize **7′** or any methylated intermediates (Fig. 2g). Then, we engineered the **7**-producing strain to express PsCXE1 downstream of a strong promoter (*PYK1* and *GPD*) and observed synthesis of **8** ($m/z^{+} = 402$, $7.2 \pm 0.5\,\mu M$ in the optimal **7**-producing strain; Supplementary Fig. 3e) with MS/MS fragmentation that matches the reported spectra of narcotolinehemiacetal (Fig. 2e and Supplementary Fig. 1f). However, due to the low efficiency in the synthesis of **8**, and the tailing of **8** on the high-performance liquid chromatography (HPLC) column, we were unable to purify **8** and conduct NMR analyses to further confirm the structure. The titre of **8** in the medium was therefore estimated by comparison to the standard curve of structurally similar **9**.

We further engineered the optimal **7**-producing strain to express PsCXE1 and PsSDR1, and observed the efficient synthesis of **9** from **8** ($m/z^{+} = 400$, $17.8 \pm 0.1\,\mu M$; Fig. 2f and Supplementary Fig. 1i). The identity of **9** synthesized in yeast was confirmed by NMR analyses (Supplementary Fig. 9 and Supplementary Table 7). The higher titre of **9** compared with **8** indicates that PsSDR1 is much more efficient than PsCXE1, and with the addition of PsSDR1, the metabolic flux is driven towards the synthesis of **9**. When PsCXE1 is removed from the **9**-producing yeast strain, the synthesis of **9** is dramatically decreased but not abolished (Supplementary Fig. 8b,d). This result, together with the structural verification of **9** synthesized from **8** suggests the structure of **8** as the cyclized narcotoline-hemiacetal, and implies that **8** is present in the absence of PsCXE1. Our results indicate that **7** is prone to cyclize and form **8** spontaneously, and the presence of PsCXE1 enhances the rate of this cyclization. As the **5**-producing strain expressing CYP82X1 also synthesized trace amounts of the $m/z^{+} = 402$ peak at RT = 2.8 min, we further introduced PsSDR1 to test if the cyclized **8** is also present in this strain. The synthesis of **9** was detected in the **5**-producing strain expressing both CYP82X1 and PsSDR1 (Supplementary Fig. 8b,d), which indicates the presence of the cyclized **8** in the $m/z^{+} = 402$ peak. Although the data indicates that the $m/z^{+} = 402$ peak at RT = 2.8 min contains the cyclized **8**, it cannot be excluded that this peak is a mixture of **6′** and **8**. The efficient synthesis of **9** from **7** catalysed by PsCXE1 and PsSDR1 indicates that **7** and **8** are also likely to be the native substrates of PsCXE1 and PsSDR1, respectively; and the 4′-O-methylation could be the final step.

**PsMT2 and PsMT3 are both required for noscapine production.** To reconstruct the full noscapine biosynthetic pathway, we engineered the **9**-producing strain to express PsMT2. However, when grown in the presence of **2** at $25\,^{\circ}C$, the engineered strain did not synthesize noscapine (Fig. 2i). Our results agree with previous *in vitro* characterization of PsMT2 (ref. 50), in which no activity was observed when incubating with **9**. We hypothesized that the activity of PsMT2 was not correctly identified in the original plant studies[37,50], and that a different O-methyltransferase enzyme may catalyse the final step in the reaction. Another O-methyltransferase (PsMT3) is encoded in the proposed gene cluster, and shares high sequence similarity with a

norcoclaurine 6-O-methyltransferase from *P. somniferum* (Ps6OMT). Previous *in vitro* characterization studies demonstrated that PsMT3 showed no activity on narcotoline and proposed that this enzyme acts as a S9OMT, similar to PsMT1 (ref. 50). In this study, we engineered the narcotoline-producing strain to express PsMT3 and did not observe conversion of **9** to noscapine in either of these strains (Fig. 2i).

Plant O-methyltransferases function as dimers in which the substrate binding pocket is formed at the interface of the two monomers[59]. While all known O-methyltransferases are characterized to act as homodimers in their native plant pathways, heterodimers formed between different O-methyltransferases have been shown to act with low efficiencies on unique substrates in a heterologous context[60]. The observation that PsMT3 or PsMT2 do not act individually to catalyse the methylation of **9**, led us to hypothesize that the synthesis of noscapine from **9** may require both O-methyltransferases. A **9**-producing strain co-expressing PsMT2 and PsMT3 efficiently converted **9** to noscapine ($m/z^+ = 414$, $14.8 \pm 0.9\,\mu M$ in the optimal **9**-producing strain; Fig. 2i and Supplementary Fig. 1j), suggesting that these methyltransferase enzymes may act as a heterodimer. The identity of noscapine was confirmed through NMR analyses (Supplementary Fig. 9 and Supplementary Table 7). Due to the high sequence similarity between Ps6OMT and PsMT3, we replaced PsMT3 with Ps6OMT in the noscapine-producing strain, and observed similar noscapine titre from the engineered yeast strains (Fig. 2i and Supplementary Fig. 10a)

The PsMT2/PsMT3 enzyme complex was then purified from *Escherichia coli* co-expressing the hexa-histidine tagged PsMT2 and T7-tagged PsMT3. The conversion of **9** to noscapine was detected when the PsMT2/PsMT3 complex was incubated with **9** and S-adenosyl methionine, while no noscapine was synthesized with purified PsMT2 or PsMT3 alone (Fig. 4a). SDS/(polyacrylamide gel electrophoresis) PAGE, western blotting and size-exclusion chromatography (SEC) analyses further support that the functional 4′-O-methyltransferase is a dimer

harbouring both PsMT2 and PsMT3 (Fig. 4b,c). The efficient conversion of noscapine from **9** suggests that the 4′-O-methylation is likely to be naturally catalysed by the PsMT2-PsMT3 heterodimer (Supplementary Fig. 10b). We also detected the synthesis of **7′** ($m/z^+ = 458$) and **8′** ($m/z^+ = 416$, $\sim 5.8 \pm 0.7\,\mu M$ in the optimal **8**-producing strain) in these strains (Fig. 2g,h and Supplementary Fig. 10b). Similar to **8**, we were unable to purify **8′** from the medium due to the low synthesis efficiency and tailing on the HPLC column, and the titre reported for **8′** was estimated by comparison to the standard curve of **1**. Since the observed activity on narcotoline for 6OMT was unexpected, we examined whether the same enzyme from a plant host that does not synthesize noscapine, *Thalictrum flavum*, would exhibit this activity (Fig. 2i). **9**-producing strains co-expressing PsMT2 and Tf6OMT did not synthesize noscapine, indicating that the observed 6OMT activity was unique to the *P. somniferum* variant. We also purified the PsMT2/Ps6OMT heterodimer from *E. coli* (Supplementary Fig. 11a,b). The *in vitro* conversion of narcotoline to noscapine is lower than that observed from the PsMT2/PsMT3 heterodimer (Supplementary Fig. 11c), which is different from the conversion efficiencies observed in yeast (Supplementary Fig. 10a). Further kinetic analysis and *in planta* characterization of these two heterodimers is required to fully understand their native functions in *P. somniferum*.

To further elucidate the ordering of the final steps in the noscapine biosynthetic pathway in yeast, we engineered both **7**- and **8**-producing strains to co-express PsMT2 and either PsMT3 or Ps6OMT (Fig. 2g,h). The narcotoline-4′-O-methyltransferase exhibits a much higher efficiency on **8** and **9** compared with **7** (Supplementary Fig. 10b). The data supports that O-methylation and dehydrogenation are more likely to be the final two steps in the biosynthesis of noscapine, and thus the full biosynthetic pathway of noscapine from **2** is proposed as shown in Fig. 1. We believe that PsMT2/PsMT3 and PsSDR1 catalyse the final two steps in yeast; however, *in planta*, the reaction matrix may be shifted due to distinct localization patterns in the plant[61–63],

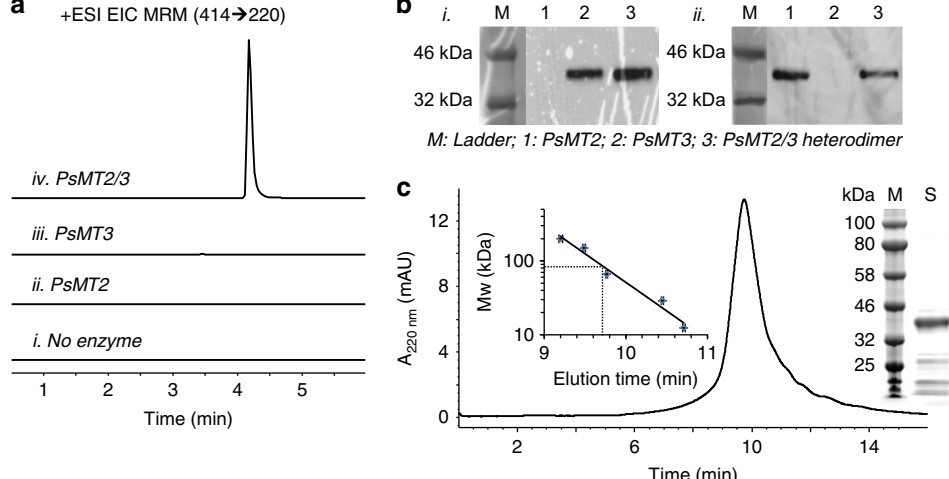

**Figure 4 | Identification of the narcotoline-4′-O-methyltransferase as the PsMT2/PsMT3 heterodimer.** (**a**) EIC MRM using noscapine's highest characteristic precursor ion/product ion transition ($414 \rightarrow 220$) of *in vitro* assays containing (i) no enzyme (control), (ii) PsMT2 homodimer, (iii) PsMT3 homodimer and (iv) PsMT2/PsMT3 heterodimer. All traces are in the same scale and are representative of three biological replicates for each enzyme. (**b**) Western blot analysis of the purified and concentrated (1) PsMT2 homodimer, (2) PsMT3 homodimer and (3) PsMT2/PsMT3 heterodimer with (i) Anti-T7 tag antibody and (ii) Anti-6X His tag antibody. Gels and blots are representative of two biological replicates. (**c**) SEC analyses of the PsMT2/PsMT3 heterodimer. The left inset is the calibration curve of the protein standards, with the RT on the *x* axis, and the molecular weight in log scale on the *y* axis. The right insert is the SDS–PAGE analysis of (M) ladder and (S) the purified and concentrated PsMT2/PsMT3 heterodimer. The molecular weight of the His-tagged PsMT2 is 40.09 kDa, the T7-tagged PsMT3 is 38.70 kDa. The calculated molecular weight of the PsMT2/PsMT3 complex is $\sim 80\,kDa$, calculated based on at least three replicates. The error bars represent the s.d. of three biological replicates.

and further physiological investigations in plant are required to understand the sequence of the final pathway steps in plant.

The extracellular titre of noscapine reaches $2.2 \pm 0.3\,\mu M$ when the yeast strain is engineered to express each enzyme on an optimized expression construct and grown in the presence of $250\,\mu M$ racemic **2** (Fig. 3d and Supplementary Data 1). The extracellular titre of noscapine was increased over sevenfold to $14.8 \pm 0.9\,\mu M$ ($\sim6\%$ overall conversion from fed **2**) by increasing the expression of CYP82X2, as suggested in the aforementioned optimization for the synthesis of **5** (Fig. 3b and Supplementary Data 1).

We then fed $125\,\mu M$ (S)- (98% purity) or (R)-(96% purity) canadine to our engineered yeast strains that synthesize the major pathway intermediates to roughly estimate the promiscuity of the pathway enzymes on each enantiomer. Interestingly, although (S)-canadine is the natural substrate of the pathway in opium poppy, (R)-canadine can also be converted through the pathway to noscapine when fed independently, but at much lower efficiencies than (S)-canadine in the early steps of the pathway (Supplementary Fig. 12, for example, $112.66 \pm 16.07\,\mu M$ versus $10.63 \pm 0.34\,\mu M$ of **4** is synthesized when fed with (S)-canadine and (R)-canadine, respectively). However, we also observed that the ratio between (S)-enantiomer and (R)-enantiomer decreases along the pathway (Supplementary Fig. 12), which may be attributed to the 4% impurity in (R)-canadine.

**Yeast biosynthesis of other protoberberine structures**. In addition to supporting a synthesis platform for noscapine and all intermediates along the noscapine biosynthetic pathway, our yeast-based platform also supports the synthesis of some other protoberberine compounds by leveraging promiscuous enzyme activities and combinations of enzymes not found in nature. For example, yeast strains engineered to express CYP82Y1 convert **2** to 1-hydroxycanadine, **10** (Supplementary Fig. 13), which is a novel protoberberine not found in nature. Yeast strains engineered to express TNMT and CYP82X2 synthesize N-methylophiocarpine, **11** (Supplementary Fig. 13). The titres of **10** and **11** are estimated by comparing to the standard curves of structurally similar **2** and **4,** respectively. Yeast strains engineered to express TNMT, CYP82Y1 and CYP82X1 synthesize **5'** (Fig. 2b

and Supplementary Fig. 1e). These off-pathway enzyme activities were optimal under different promoters from those for the native pathway activities (Supplementary Fig. 3b,f,g).

**Yeast biosynthesis of noscapine from norlaudanosoline**. Finally, we examined extension of the noscapine biosynthetic pathway in yeast to support biosynthesis from a less expensive fed precursor. The biosynthesis of **2** from an early benzylisoquinoline alkaloid (BIA) intermediate, **12**, has recently been reported in *S. cerevisiae*[53]. Our successful reconstitution and optimization of the biosynthetic pathway from **2** to noscapine enables the connection of these two pathway parts to achieve the microbial biosynthesis of noscapine from **12**. Since Ps6OMT can function together with PsMT2 as the narcotoline-4'-O-methyltransferase, yeast strains were engineered to express AtATR1, Ps6OMT, 3'-hydroxy-N-methylcoclaurine-4'-O-methyltransferase from *P. somniferum* (Ps4'OMT), coclaurine N-methyltransferase from *P. somniferum* (PsCNMT), berberine bridge enzyme from *P. somniferum* (PsBBE), S9OMT from *P. somniferum* (PsS9OMT or PsMT1), CAS from *Coptis japonica* (CjCAS), PsTNMT, CYP82Y1, CYP82X2, PsAT1, CYP82X1, PsCXE1, PsSDR1 and PsMT2 without or with PsMT3 (CSYN15 without PsMT3, CSYN16 with PsMT3, Supplementary Data 1). When grown in the presence of 2 mM racemic **12**, which is assumed to contain equal amounts of each enantiomer, CSYN15 synthesized $0.048 \pm 0.011\,\mu M$ of noscapine, while CSYN16 synthesized a significantly higher level of noscapine ($0.49 \pm 0.16\,\mu M$, paired $t$-test, $P = 0.000083$; Fig. 5a,b). CYP82X2 and PsS9OMT are enzymes associated with two bottleneck points in the pathway. By increasing the expression levels of both CYP82X2 and PsS9OMT, the synthesis of noscapine was increased over threefold ($1.64 \pm 0.38\,\mu M$) to that observed from CSYN16 (paired $t$-test, $P = 0.00047$; Fig. 5b,c and Supplementary Data 1). To estimate the transport efficiency of the pathway metabolites, intracellular concentrations of major pathway intermediates were estimated and compared with the concentrations measured in the growth media (Supplementary Fig. 14). Our study indicates that with the exception of **5**, all major pathway intermediates are efficiently transported out of yeast cells to the medium, with the export of **6** and **7** relatively lower.

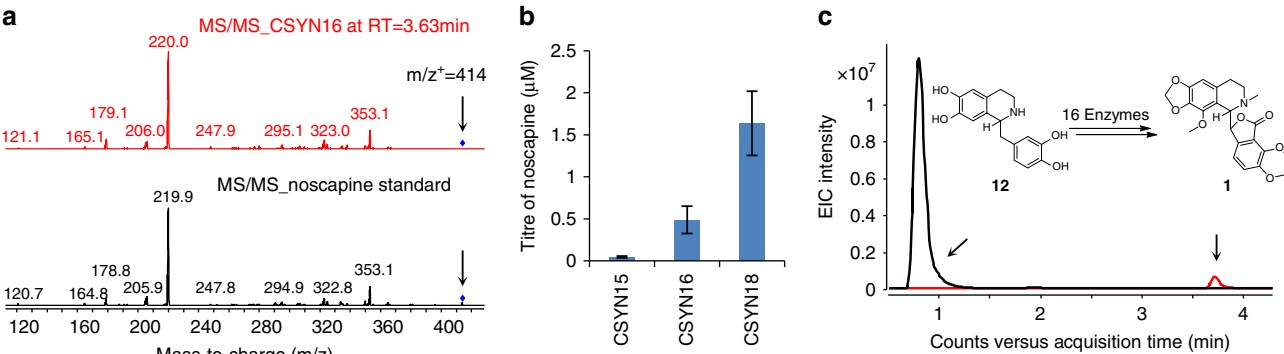

**Figure 5 | LC–MS analysis of yeast strains synthesizing noscapine from norlaudanosoline.** (**a**) MS/MS spectra of noscapine standard (black) and medium of the noscapine-producing strain CSYN16 (Supplementary Data 1; red). (**b**) Noscapine titre analysed from engineered strains harbouring different sets of expression cassettes. CSYN15: AtATR1, Ps6OMT, Ps4'OMT, PsCNMT, PsBBE, TfS9OMT, CjCAS, CYP82Y1, CYP82X2, PsAT1, PsSDR1, PsTNMT and PsMT2 expressed from the chromosome; CYP82X1 and PsCXE1 expressed from a low-copy plasmid (Supplementary Data 1). CSYN16: AtATR1, Ps6OMT, Ps4'OMT, PsCNMT, PsBBE, PsS9OMT, CjCAS, CYP82Y1, CYP82X2, PsAT1, PsSDR1, PsTNMT, PsMT2, CYP82X1, PsCXE1 and PsMT3 expressed from the chromosome (Supplementary Data 1). CSYN18: AtATR1, Ps6OMT, Ps4'OMT, PsCNMT, PsBBE, TfS9OMT, CjCAS, CYP82Y1, CYP82X2, PsAT1, PsSDR1, PsMT2, PsTNMT, CYP82X1, PsCXE1 and PsMT3 expressed from the chromosome; CYP82X2, PsS9OMT expressed from a low-copy plasmid (Supplementary Data 1). (**c**) EIC of $m/z^+ = 288$ (black) and 414 (red) of noscapine-producing yeast strain CSYN18. Bars represent mean values $\pm 1$ s.d. of five biological replicates, and the error bars represent the s.d. of the replicates.

## Discussion

The 10-gene cluster encoding enzymes responsible for the conversion of scoulerine to noscapine was first discovered and verified in *P. somniferum* in 2012, and activities of many of the enzymes were proposed based on virus-induced gene silencing assays in the native plant host[37]. A series of *in vitro* characterization studies on the biosynthetic enzymes involved in the synthesis of noscapine were recently published[47–50], which provide additional insights into the biosynthesis of noscapine in opium poppy. In this work, we reconstituted the noscapine gene cluster in yeast to achieve the microbial production of secoberberine (5′ and 7) and phthalideisoquinoline (noscapine and 9) alkaloids from fed 2 or 12. The activities of enzymes encoded in the gene cluster reconstituted in yeast agree well with the *in planta* and *in vitro* biochemical characterization studies, and provides additional insights into the biosynthesis of noscapine. Our work also enables the supply of many pathway intermediates and derivatives for structural confirmation and further investigation.

We primarily fed racemic canadine, 2, to our strains, which is assumed to be composed of equal amounts of each enantiomer. The substrate specificity of the noscapine pathway enzymes towards different enantiomers has not previously been investigated[47,48,56]; instead, racemic canadine has typically been used to characterize the enzymes along the pathway *in vitro*. In addition, individual enantiomer standards of pathway intermediates between canadine and noscapine are unavailable. Thus, it is difficult to estimate the ratio between each enantiomer along the noscapine pathway. We fed (S)- and (R)-canadine to our engineered yeast strains, and the data indicated that although our racemic canadine-fed strains mainly produce (S)-enantiomers of the pathway metabolites, (R)-enantiomers may also be present in small quantities. However, considering that the (R)-canadine is of 96% purity, it is difficult to estimate the exact percentage of pure (R)-enantiomers in strains fed with (R)-canadine. In addition, when fed with racemic 2, the conversion of (S)- and (R)-canadine to the pathway intermediates may shift from when they are fed separately to the engineered yeast strains, and whether or not (R)-canadine can be converted when both enantiomers are present is unknown.

Recent advances in the *de novo* production of reticuline in yeast[64,65] highlight the possibility of constructing yeast strains that can synthesize noscapine and related metabolites from simple carbon and nitrogen sources versus feeding of expensive intermediate alkaloids. Whereas the microbial platforms described here are suitable for smaller scale production of noscapine and related compounds, for example, for drug screening and enzyme activity identification, the *de novo* production of noscapine and related molecules could provide a cost-efficient process for the production of these valuable molecules. However, no less than eleven enzymes, including four cytochrome P450s, will need to be introduced into the reticuline-producing strain for the *de novo* synthesis of noscapine. The full *de novo* pathway to noscapine is thus composed of up to 18 heterologous enzymatic steps from tyrosine and at least 23 different heterologous enzymes, including 5 cytochrome P450s. Our data indicates that as the heterologous pathway introduced into yeast includes more P450s, the conversion efficiency of each P450 will decrease, which may be due to challenges associated with endoplasmic reticulum loading and cofactor regeneration. Therefore, introducing a pathway comprising the expression of at least 5 cytochrome P450s will require further engineering of the yeast background to increase efficiencies. In addition, the reported strains produce relatively low levels of reticuline from tyrosine with titres reported at $< 100 \mu g l^{-1}$ (refs 64,65). Thus, further optimization will be required to develop improved reticuline-producing platform strains to ultimately drive the metabolite flux from reticuline to noscapine. To develop a commercially competitive process for noscapine production via microbial fermentation, titres would need to reach at least the range of 10–20 mM (ref. 6). In addition to improving the pathway within a single yeast strain to increase production levels of noscapine[8,64,65], other engineering strategies such as co-culturing a noscapine-producing strain with a canadine- or norcoclaurine-producing strain may also be a promising strategy. Multiple-strain biosynthetic strategies have recently been described to achieve enhanced microbial production of taxanes and opioids[9,11].

Throughout our study, we measured the pathway metabolite levels in the growth medium from the engineered yeast strains. Our data indicates that most metabolites were transported out of the yeast cells efficiently (Supplementary Fig. 14). However, the export of 5, 6 and 7 are not as efficient, and thus the titre in the growth medium can be an underestimate of the production of the corresponding molecules. The efficient export of the other pathway intermediates may lead to lower intracellular concentrations, and can be deleterious to the metabolic flux through the pathway especially for slower enzymatic steps. Future engineering efforts directed to optimizing the export of pathway metabolites and noscapine may be critical to further enhancing the production of target molecules (for example, noscapine or target pathway intermediates).

We observed that PsSDR1 or PsMT2/PsMT3 could catalyse the final step in the synthesis of noscapine in yeast (Fig. 1). Previous *in planta* and *in vitro* studies implied that the synthesis of 7 to noscapine is likely to be branched into several routes[47]. Our work validates the reaction matrix of the final three steps (that is, PsCXE1, PsSDR1 and PsMT2/PsMT3), and showed the efficiency of each route in yeast. While the noscapine pathway in plants may follow the sequence of the final three steps proposed based on our yeast pathway reconstitution study, it is also possible that the pathway exhibits slightly different patterns in the native plant host. For example, the pathway enzymes have specific cellular localization and spatial organization in the opium poppy that may result in the biosynthetic matrix being routed in different patterns depending on the pathway localization in the plant. For example, alkaloid synthesis in opium poppy involves several cell types and transport steps[61]. Therefore, the pathway organization and sequence is much influenced by the subcellular localization of pathway enzymes. The previous *in planta* study found that the enzymes catalysing the final three steps of morphine biosynthesis are among the most abundant active proteins in the latex sub-proteome, which indicates that the final three steps are likely to principally occur in the laticifers[61]. Among the noscapine pathway enzymes, only PsSDR1 is enriched in laticifers[49]. Thus, biosynthesis of noscapine may follow similar organization to morphine, with PsSDR1 catalysing the final step in laticifers. Additional investigations in opium poppy are necessary to further understand the organization and sequence of noscapine biosynthesis. More broadly, the microenvironments in yeast, *in vitro* and *in planta* in which the pathway enzymes function can be different, and these differences can result in distinct substrate preferences, enzyme activities and product transportation processes. Thus, observations of pathway sequence and efficiency across these different contexts should ultimately be considered with regard to these differences.

Our study also uncovered a unique O-methylation step that is likely to be natively catalysed by an O-methyltransferase heterodimer in opium poppy. While previous studies demonstrated that plant O-methyltransferase heterodimers can exhibit different substrate specificity from the corresponding homodimers in synthetic contexts[59,60], our results suggest a role for O-methyltransferase heterodimers in native plant pathways.

These findings suggest that some uncharacterized O-methylation steps in the biosynthesis of plant natural products may be catalysed by heterodimers rather than homodimers. In addition, our results suggest that O-methyltransferases can catalyse more than one methylation reaction in their original plant hosts by acting on one substrate as a homodimer and acting on different substrates when forming heterodimers with different O-methyltransferase partners. Further *in planta* investigations are essential for validating that PsMT2 and PsMT3 form a heterodimer in opium poppy, and studying the native functions of PsMT2, PsMT3, Ps6OMT and their dimers.

Step-wise pathway reconstitution in yeast can be applied to bridge *in planta* and *in vitro* characterizations of plant natural product pathways to provide a complementary approach for better understanding and engineering the biosynthetic potential of plants. The yeast-based plant pathway reconstitution also has the advantage of simultaneously building production strains for potentially valuable compounds, including all intermediates along a pathway and novel derivatives thereof. While the end-product noscapine accumulates in plants at low levels, most of the pathway intermediates, which can serve as interesting lead compounds for drug discovery, are in exceedingly low abundance in the producing plants, and thus not accessible. In addition, enzymatic steps can be readily removed or introduced in the yeast platform to enable the synthesis of novel structures such as **10**. Our work demonstrates the potential of synthetic biology as a powerful tool for aiding complex pathway elucidation, while enabling microbial production of valuable natural or unnatural medicinal compounds originated from plants.

## Methods

**Compounds and strain culture condition.** Racemic canadine (98% purity), **2**, norlaudanosoline (98% purity), **12**, and reticuline were purchased from Finetech Industry Limited, Santa Cruz Biotechnology and Specs, respectively. (*R*)- and (*S*)-enantiomers of **2** were purchased from Toronto Research Chemicals. S-Adenosyl methionine (80% purity) and noscapine (97% purity), **1**, were purchased from Sigma-Aldrich. Narcotoline (>95% purity), **9**, was purified from **9**-producing yeast strain in this study as described below. For DNA manipulation and amplification, we used chemically competent or electrocompetent *E. coli* strains TOP10 and ccdB Survival 2 T1$^R$ purchased from Life Technologies. *E. coli* strains were grown at 37 °C in lysogeny broth medium obtained from Fisher Scientific supplemented with 100 mg ml$^{-1}$ ampicillin (EMD Chemicals) or 50 mg ml$^{-1}$ kanamycin (Fisher Scientific) for plasmid maintenance. All engineered yeast strains described in this work, as listed in Supplementary Data 1, were constructed in a haploid W303α background (MATα *leu2-3,112 trp1-1 can1-100 ura3-1 ade2-1 his3-11,15*) [66]. Yeast strains were cultured at 25 or 30 °C in complex yeast extract peptone dextrose (YPD, all components from BD Diagnostics) medium or SDM containing yeast nitrogen base (YNB) without Amino Acids (BD Diagnostics), ammonium sulfate (Fisher Scientific), 2% dextrose and the appropriate dropout solution for plasmid maintenance. 200 mg l$^{-1}$ G418 sulfate (Calbiochem) or 200 mg l$^{-1}$ Hygromycin B (Life Technologies) were used in YPD medium for selection. BL21(DE3) and Ni-NTA agorose resin used for *E. coli* protein expression and hexa-histidine tagged protein purification were kindly provided by Professor Chaitan Khosla, and T7 Tag Affinity Purification Kit used for T7-tagged protein purification was purchased from EMD Millipore. NuPAGE Novex 4–12% Bis-Tris Protein Gels (1.0 mm, 10 well) used for purified protein SDS–PAGE analysis and western blot were purchased from Life Technologies and Color Prestained Protein Standard (Broad Range, 11–245 kDa) used as the protein marker was obtained from New England Biolabs. The Anti-6X His tag antibody (horseradish peroxidase (HRP), ab1269) and Anti-T7 tag antibody (HRP, ab21477) used for western blot of PsMT2/PsMT3 and PsMT2/Ps6OMT heterodimers were purchased from Abcam PLC. Size-exclusion chromatography was performed on a BioSep-SEC-s4000 column (Phenomenex, kindly provided by Professor Chaitan Khosla), and the column was equilibrated with Gel Filtration Markers Kit for Protein Molecular Weights 12,000–200,000 Da purchased from Sigma.

**General techniques for DNA manipulation.** Plasmid DNA was prepared using the QIAprep Spin Miniprep Kit (QIAGEN) and Econospin columns (Epoch Life Science) according to manufacturer's protocols. PCR reactions were performed with Taq polymerase, PfuUltra Hotstart DNA Polymerase (Agilent Technologies) and Expand High Fidelity PCR System (Roche Life Science) according to manuacturer's protocols. PCR products were purified by Zymoclean Gel DNA Recovery Kit (Zymo Research). All DNA constructs were confirmed through DNA

sequencing by Elim Biopharmaceuticals Inc. Restriction enzymes (NEB) and T4 ligase (NEB) were used to digest and ligate the DNA fragments, respectively. BP Clonase II Enzyme Mix, Gateway pDONR221 Vector and LR Clonase II Enzyme Mix (Life Technologies) and the *S. cerevisiae* Advanced Gateway Destination Vector Kit (Addgene)[67] were used to perform Gateway Cloning. Gibson one-pot, isothermal DNA assembly[68] was conducted at 10 μl scale by incubating T5 exonuclease (NEB), Phusion polymerase (NEB), Taq ligase (NEB) and 50 ng of each DNA fragment at 50 °C for 1 h to assemble multiple DNA fragments into one circular plasmid. Yeast strains are constructed through homologous recombination and DNA assembly[23]. Yeast strains and plasmids utilized in this study are listed in Supplementary Data 1 and 2. DNA sequences of genes involved in this work are listed in Supplementary Data 3.

**Plasmids and yeast construction.** Genes encoding CYP82Y1, CYP82X2, PsAT1, CYP82X1, PsCXE1, PsSDR1, PsMT2 and PsMT3 (*CYP82Y1*, *CYP82X2*, *PsAT1*, *CYP82X1*, *PsCXE1*, *PsSDR1*, *PsMT2* and *PsMT3*) were codon-optimized for expression in *S. cerevisiae* using the GeneArt GeneOptimizer programme (Life Technologies) and synthesized by GeneArt (Life Technologies) or integrated DNA technologies (IDT). The synthesized genes and *Ps6OMT* and *Tf6OMT* (amplified from previous constructs[52]) were introduced into Gateway pDONR221 using BP Clonase II Enzyme Mix to yield pCS2575, 2809, 3115-3120 and 3215 (Supplementary Data 2). The genes were subsequently recombined into selected pAG expression vectors from the *S. cerevisiae* Advanced Gateway Destination Vector Kit[67] using LR Clonase II to generate the corresponding pAG yeast expression constructs pCS3152–3154, 3158–3159, 3161, 3164–3165, 3168–3169, 3173–3174, 3177, 3179 and 3619 (Supplementary Data 1). *PsAT1* was amplified with primer PsAT1-F (5′-caccagaacttagtttcgacgg attctagaactagttatacaatggctaccatgt catctgctgc-3′) and PsAT1-R (5′-attacatgactcgaggtcgacggtatcgat aagcttttagaataattgc aagatttctctcaagtggtgttc-3′) and inserted into pAG415GPD-ccdB digested with SacI and HindIII with yeast *in vivo* homologous recombination to yield pCS3124.

Construction of new destination vectors for Gateway cloning. The *HXT7* promoter was cloned from pCS2658 with primers HXT7p-SacI-F (5′-aaaaaagagc tcctcgtaggaacaatttcgggccc-3′) and HXT7p-SpeI-R (5′-aaaaaaactagtgtttttttgattaaaatta aaaaaacttttt gttttttgtgtttattcttttgttc-3′). The resulting DNA fragment was digested with SacI and SpeI and inserted in pAG413GPD-ccdB, pAG414GPD-ccdB and pAG416GPD-ccdB (digested with SacI and SpeI) to yield pAG413HXT7-ccdB (pCS3111), pAG414HXT7-ccdB (pCS3112) and pAG416HXT7-ccdB (pCS3113) (Supplementary Data 2). pCS3116 was recombined into pCS3111, pCS3112 and 3113 using LR Clonase II to generate pCS3172, pCS3151 and pCS3191, respectively (Supplementary Data 1). The *ccdB-Cam$^R$* cassette was amplified from pAG413GPD-ccdB by ccdB-gibson-F (5′-gggaacaaaagctggagctcccgcggcccatcacaagt ttgtacaaaaaagctgaac-3′) and ccdB-gibson-R (5′-ctatagggcgaattgggtaccgtttaaacgcggc cgccccatcaccacttttgtacaagaaagctg-3′). The DNA fragment was inserted into vectors amplified through PCR from pAG413GPD-ccdB, pAG414GPD-ccdB, pAG415GPD-ccdB and pAG416GPD-ccdB by pAG-gibson-F (5′-gagctccagctttttgt tcccttttag-3′) and pAG-gibson-R (5′-gtacccaattcgccctatagtgagtcg-3′) using Gibson assembly to yield pAG413-ccdB (pCS2811), pAG414-ccdB (pCS2812), pAG415-ccdB (pCS2813) and pAG416-ccdB (pCS2814), respectively (Supplementary Data 2). pCS3141 (Supplementary Data 2) was recombined into pCS2812 using LR Clonase II to generate pCS3607 (Supplementary Data 1).

Construction of N-terminus mutants of CYP82Y1. Using pCS2809 as the parent vector, the N-terminus domain of CYP82Y1 was swapped with the N-terminus domain of (S)-cis-N-methylstylopine-1-hydroxylase (MSH), lanosterol 14-demethylase from *S. cerevisiae* (L14D), a cytochrome P450 reductase partner CPR from *Candida maltose* (CmCPR) and cheilanthifoline synthase from *E. californica* (EcCFS) using Gibson assembly to yield pCS2842, and 3121-3123 encoding CYP82Y1A, CYP82Y1B, CYP82Y1C and CYP82Y1D (Supplementary Data 2 and 3), respectively. The primers used for constructing the N-terminus mutants of CYP82Y1 include pENTR_CYP82Y1-F (5′-cctgaagctagtggtgcttggc-3′) and pENTR_CYP82Y1-R (5′-tgtatagaagcctgctttttttgta caaagttgg-3′) for amplifying the vector including truncated CYP82Y1, MSH-Ntag-F (5′-aaaaaagcaggcttctatacaatgagaaccgaatccatcaagaccaac-3′) and MSH-Ntag-R (5′-caagcacc actagcttcaggagccaatttcttagttgggtttcttctacc-3′) for amplifying the MSH N-terminus tag, L14D-Ntag-F (5′-aaaaaagcaggcttctatacaatgtctgctaccaagtcaatcgttgg-3′) and L14D-Ntag-R (5′-caagcaccactagcttcaggacggtcctttctcaaagaatatagtaattgcc-3′) for amplifying the L14D N-terminus tag, CmCPR-Ntag-F (5′-aaaaagcaggcttctatacaatggctttggataaattggatttgtatgtt attattg-3′) and CmCPR-Ntag-R (5′-caagcaccactagcttcaggatcttgtggttgatccaagaattgatttta gc-3′) for amplifying the CmCPR N-terminus tag, and EcCFS-Ntag-F (5′-aaaaaagcaggctt ctatacaatggaagagtctttatgggtcgttactg-3′) and EcCFS-Ntag-R (5′-caagcaccactagcttcaggccatt ccatagttgagatag atgatgatttctttaac-3′) for amplifying the EcCFS N-terminus tag. The mutant genes were subsequently recombined into pAG413GAL-ccdB from the *S. cerevisiae* Advanced Gateway Destination Vector Kit using LR Clonase II to generate pCS2844 and 3180–3182 (Supplementary Data 1). Similarly, pCS2842 was also recombined into pAG413GPD-ccdB to generate pCS3166 (Supplementary Data 1).

Construction of P450 expression plasmids with different promoters. *CYP82Y1A* was inserted into pCS2657–2661 and 2663–2664 (ref. 69) using Gibson assembly to yield pCS3125-3131 (Supplementary Data 2). *CYP82X1* was inserted into pCS2658–2660 and 2663 using Gibson assembly to yield pCS3132–3135 (Supplementary Data 2). *CYP82X2* was inserted into pCS2659 and 2663 using Gibson assembly to yield pCS3136 and 3137 (Supplementary Data 2). These promoter-gene-terminator cassettes of CYP82Y1A, CYP82X2 and CYP82X1

flanked by *attL* recombination sites were subsequently recombined into pCS2811, pCS2812 and pCS2814, using LR Clonase II to generate pCS3136-3137, 3149–3150, 3156, 3167, 3171, 3175, 3178, 3183–3187 and 3188–3189 (Supplementary Data 1). *CYP82Y1* and *CYP82Y1A* were also recombined into pAG413GPD-ccdB and pAG413GAL-ccdB, *CYP82X1* recombined into pAG416GPD-ccdB and pAG416GAL-ccdB, and *CYP82X2* recombined into pAG414GAL-ccdB, pAG414GPD-ccdB, pAG413HXT7-ccdB (pCS3111), pAG414HXT7-ccdB (pCS3112) and pAG416HXT7-ccdB (pCS3113) to construct expression cassettes regulated by the *GPD* promoter and *HXT7* promoter (pCS3151, 3165, 3168–3169, 3172–3174, 3179 and 3191; Supplementary Data 1).

Construction of *in vitro* expression plasmids. *PsMT2*, *PsMT3* and *Ps6OMT* were each amplified with primers PsMT2-pET-F (5′-aattttgtttaactttaa gaaggagatataatgga aatccacttggaatcccaagaac-3′) and R (5′-atctcagtggtggtggtggtggtgctcgagtgggtaagcaacga taatagatggcatg-3′), and PsMT3-pET-F (5′-tttgtttaactttaagaaggagatatacatatggaagttgtc tccaa gatcgatcaag-3′) and R (5′-catttgctgtccaccagtcatgctagccatgtgtggataagcttcaataacg gattgcaag-3′), and Ps6OMT-pET-F (5′-tttgtttaactttaagaaggagatatacatatggaaacagtaag caagattgatcaacaaaacc-3′) and R (5′-catttgctgtccaccagtcatgctagccatataaagggtaagcctcaat tacagattggacag-3′), respectively, to generate DNA fragments encoding PsMT2 with hexa-histidine tag, PsMT3 with T7 tag and Ps6OMT with T7 tag at the C terminus. The PsMT2 fragment was then inserted into pET28a (amplified by pET28_gibson_F (5′-tat atctccttccttaaagttaaacaaaattatttctagagggg-3′) and pET28 gibson R (5′-tgcttacccactcgagcac caccaccacc-3′)) using Gibson assembly to construct pCS3306 (Supplementary Data 2). The PsMT3 and Ps6OMT fragments were inserted into pET21a (amplified by pET21_gibson_F (5′-atgtatatctccttcttaaagttaaacaaaattatttctagagg-3′) and pET21 gibson R (5′-atggctagcatgactggtggacagcaaatgggttgacgaattcgagctccgtcgacaag-3′)) using Gibson assembly to construct pCS3307 and pCS3534, respectively (Supplementary Data 2).

Plasmid- and chromosome-based DNA assembly. *Ps6OMT*, *Ps4'OMT*, *PsBBE*, *PsMT1*, *CjCAS*, *PsTNMT*, *PsAT1*, *PsCXE1*, *PsSDR1*, *PsMT2* and *PsMT3* were each inserted into pCS2663, pCS2661, pCS2663, pCS2657, pCS2656, pCS2657, pCS2656, pCS2664, pCS2661, pCS2663 and pCS2656, respectively, using Gibson assembly to yield pCS3138, 2803, 3060 (ref. 53), 3064 (ref. 53), 3075 (ref. 53) and 3139–3144, respectively (Supplementary Data 2). For plasmid-based DNA assembly, certain expression cassettes were PCR amplified and incorporated into pYES1L to yield pCS3155, 3162–3163 and 3619 (Supplementary Data 1). For chromosome-based DNA assembly, 500–1,000 bp flanking the target loci (from genomic DNA), certain expression cassettes (from pCS2803, 3060, 3064, 3075, 3126, 3132, 3134, 3139–3144 and 3147; Supplementary Data 2), and selection marker (*KanMX*, *leu2*, *Hygro*[R] from pCS1056, 3145-3147) were PCR amplified and incorporated into selected *loci* to generate CSY1000 and CSY1054 (Supplementary Data 1). Both plasmid- and chromosome-based DNA assemblies were conducted through transformation of yeast with a mixture of 100 ng of each DNA fragment using electroporation; and the correctly assembled constructs were verified through PCR analysis of the junctions between each adjacent DNA fragment[23,69].

Single-gene chromosomal integration. The AtATR1 integration cassette was PCR amplified from pCS1056 (ref. 52) with URA3-pUG-F (5′-ggtatatatacgcata tgtggtgttgaagaaacatgaaattgcccagtattcttaacccaactgcacagaacaaaaaacctgcaggaaacgaagat aaatcatgagattgtactgagagtgcac-3′) and URA3-pUG-R (5′-atcattacgaccgagattcccggg taataactgatataataattaaattgaagctctaatttgtgagtttagtatacatgcatttacttataatacagttttttacgactcac tatagggagacc-3′). The resulting DNA fragment was transformed into W303α by standard lithium acetate transformation, and selected from YPD medium supplemented with G418. The integration event was confirmed by PCR screening and functional verification. The integration selection marker was rescued by heterologous expression of Cre recombinase[70]. The P$_{GPD}$-*PsTNMT*-T$_{CYC1}$ cassette was PCR amplified from pCS3139 by GPD-pCS1056-F (5′-tgtactgagagtgcaccata tggtttaa acgagctcttcgagtttatcattatcaatactcgc-3′) and GPD-pCS1056-R (5′-tatacgaag ttatatattaagggttgtcgacgc ggccgcaaattaaagccttcg-3′), and inserted into pCS1056 and PCR amplified by pCS1056-F (5′-gccggccgcgtgcgacaacccttaatataacttctgataatgtatg-3′) and pCS1056-R (5′-gtttaaaccatatggtgcactctcagtacaatctgc-3′) using Gibson assembly to yield pCS3145 (Supplementary Data 2). P$_{HXT7}$-*CYP82Y1A*-T$_{CYC1}$ (pCS3146), P$_{HXT7}$-*CYP82X2*-T$_{CYC1}$ (pCS3147) and P$_{GPD}$-*PsAT1*-T$_{CYC1}$ (pCS3148) integration cassettes were constructed using a similar procedure (Supplementary Data 2). P$_{GPD}$-*PsTNMT*-T$_{CYC1}$, P$_{HXT7}$-*CYP82X2*-T$_{CYC1}$ and P$_{GPD}$-*PsAT1*-T$_{CYC1}$ were then integrated into CSY996 strain sequentially to yield CSY997, CSY998 and CSY999 (Supplementary Data 1). The primers used to amplifying these expression cassettes for integration into different *loci* include HIS3 pUG F (5′-attggcattatcacataatgaat tatacattatatataaagtaatgtgatttcttcgaagaatatactaaaaaatgagcaggcaagataaacgaaggcaaagatg agattgtactgagagtgcac-3′) and HIS3 pUG R (5′-agtatcatactgttcgttatacatactttactgacattc ataggtatacatatatacacatgtatatatatctgatgctgcagcttttaaaatcggtgtcactacgactcactatagg gacc-3′), Leu2 pUG F (5′-ttttccaataggtggttagcaatcgtcttactttctaactttctttacctttttacatttca gcaatatatatatatatttcaaggatatacattctaatgagattgtactgagagtgcacc-3′) and Leu2 pUG R (5′-tacccctatgaacatattccattttgtaatttcgtgttcgtttctattatgaatttcatttataaagttttatgtacaaatat cataaaaaaagagaatctttttacgactcactatagggagacc-3′), and Trp1 pUG F (5′-gtatacgtgatta agcacacaaaggcagcttggagtatgtctgttattaatttcacaggtagttctggtccattggtgaaaagattgtactgagag tgcac-3′) and Trp1 pUG R (5′-tatgtttagatctttttatgcttgctttttcaaaaggcctgcaggcaagtgcac aaacaatactTaaataaataatactcagcgactcactatagggagacc-3′).

### Culture conditions for metabolite production.
For all the assays reported in this work, yeast were grown in 500 μl media in 96-well plates (BD falcon) covered with AeraSeal film (Excel Scientific), at 480 r.p.m., 80% humidity in a Kuhner Lab-Therm LX-T plate shaker. For step-wise reconstitution of the noscapine

biosynthetic pathway from canadine, yeast strains were first cultured overnight in 500 μl SDM with 2% dextrose. 5–10 μl of the overnight seed culture was inoculated in fresh 500 μl SDM with 250 μM canadine and 2% galactose, and incubated at 25 or 30 °C for 72 h before metabolite analysis of the growth medium. For the synthesis of noscapine from canadine and norlaudanosoline, 5–10 μl of the overnight seed culture was inoculated in 500 μl 5 × concentrated SDM (10% dextrose, 5 × YNB) and incubated at 25 °C for 48 h (ref. 54). The cell were then harvested by centrifugation at 15,627*g* for 5 min, pellets resuspended in fresh SDM with 2 mM norlaudanosoline and 2% galactose, and incubated at 25 °C for 96 h before metabolite analysis[54].

### Analysis of benzylisoquinoline alkaloid production.
For all assays, yeast culture medium was analysed by reverse phase LC–MS/MS on an Agilent 6300 Series Ion Trap LC–MS (Agilent ZORBAX SB-Aq 4.6 × 50 mm, 5 μm) and an Agilent 6420 triple quad LC–MS (Agilent EclipsePlus C18, 2.1 × 50 mm, 1.8 μm) using positive ionization. On the Agilent 6300 Series Ion Trap LC–MS, metabolites in the medium were separated on a linear gradient of 20% CH$_3$OH (v/v in water, 0.1% acetic acid) to 60% CH$_3$OH (v/v in water, 0.1% acetic acid) over 7 min with a flow rate of 0.5 ml min$^{-1}$. On the Agilent 6420 triple quad LC–MS, metabolites in the medium were separated on a linear gradient of 10% CH$_3$CN (v/v in water, 0.1% formic acid) to 50% CH$_3$CN (v/v in water, 0.1% formic acid) over 5 min with a flow rate of 0.4 ml min$^{-1}$ for scan mode; or 10% CH$_3$CN (v/v in water, 0.1% formic acid) to 40% CH$_3$CN (v/v in water, 0.1% formic acid) over 5 min with a flow rate of 0.4 ml min$^{-1}$ for multiple reaction monitoring (MRM) mode. The identity of each compound was verified by comparing MS/MS spectra of samples and literature or standards. The conversion of each enzymatic step was estimated by comparing the integrated peak area of extract ion chromatograms (EICs) to a standard curve. The quantities of **3**, **4**, **5**, **5′**, **6**, **7** and **9** were estimated by comparing to the stand curves of the corresponding purified and NMR verified standards. The quantities of **8**, **8′**, **10** and **11** were estimated by comparing to the standard curves of structurally similar **9**, **1**, **2** and **4**, respectively, due to the lack of standards. Noscapine levels were quantified by MRM using the highest characteristic precursor ion/product ion transition (Supplementary Table 8), and comparing the integrated peak area to a standard curve of noscapine.

### Estimation of metabolite intracellular concentrations.
To estimate the intracellular concentrations of the pathway metabolites, 500 μl overnight seed culture of CSYN16 was first inoculated in 50 ml 5 × concentrated SDM (10% dextrose and 5 × YNB) and incubated at 25 °C for 48 h (ref. 54). The cells were harvested by centrifugation at 11,305*g* for 5 min, pellets resuspended in fresh SDM with 2 mM norlaudanosoline and 2% galactose, and incubated at 25 °C (ref. 54). Three 1 ml cell cultures were collected each day after the induction until day four. The yeast culture medium of each 1 ml sample was separated from cell pellets by centrifugation at 15,871*g* for 2 min. The cell pellets were extracted with 500 μl acetone. The organic phase was separated, evaporated to dryness and dissolved in 10% CH$_3$CN (v/v in water, 0.1% formic acid) in a volume equal to that of the cell pellet (normally range 30–40 μl). The medium and cell pellet extracts were analysed on an Agilent 6420 triple quad LC–MS (Agilent EclipsePlus C18, 2.1 × 50 mm, 1.8 μm) using positive ionization. The substrate and pathway metabolites reticuline, scoulerine, tetrahydrocolumbamine, **1–9** and **12** were monitored by MRM using the highest characteristic precursor ion/product ion transition (Supplementary Table 8). The quantities of **1–7**, **9** and **12** were estimated by comparing to the standard curves of corresponding purified or purchased standards. Due to the lack of the corresponding authentic standards, the quantities of scoulerine and tetra-hydrocolumbamine were estimated using the structurally similar **2** as a standard, based on the fact that similar structures usually exhibit similar ionization efficiency. **8** was not quantified as the concentration of **8** is below the detection limit.

### Compound purification and structural elucidation.
To obtain sufficient amount of **1**, **3**, **4**, **5**, **5′**, **6**, **7** and **9** for structural elucidation by NMR, and for *in vitro* characterization of the PsMT2/PsMT3 heterodimer, the 500 μl *in vivo* assay was scaled up to 200–1,000 ml with each 100 ml of medium containing 2% galactose and 12 mg canadine, and cultured in 500 ml shake flask at 25 °C for 120 h. Yeast culture medium was separated from cell pellets by centrifugation at 4,416*g* at 5 min. The supernatant was incubated with 20 g of Amberlite XAD4 (Sigma-Aldrich) overnight. XAD4 resins were then collected through vacuum filtration, and metabolites were recovered from the resin by extracting with 50 ml MeOH for three times. The resultant organic extracts were combined and evaporated to dryness, redissolved in 20% CH$_3$CN (v/v in water, 0.1% formic acid). **1**, **3**, **4**, **5′**, **6** and **7** were purified by reverse-phase HPLC (Varian Pursuit XRs-C18, 5 μm, 250 × 10 mm) on an isocratic elution with 20% CH$_3$CN (v/v) over 15 min and 95% CH$_3$CN (v/v) for 5 min in water (0.1% formic acid) at a flow rate of 4 ml min$^{-1}$; and **5** and **9** were purified by reverse-phase HPLC (Varian Pursuit XRs-C18, 5 μm, 250 × 10 mm) on an isocratic elution with 15% CH$_3$CN (v/v) over 15 min and 95% CH$_3$CN (v/v) for 5 min in water (0.1% formic acid) at a flow rate of 4 ml min$^{-1}$. The $^1$H, $^1$H–$^1$H correlation (COSY), heteronuclear multiple quantum coherence (HMQC) and heteronuclear multiple bond correlation (HMBC) NMR spectra of **6** and **7**, the $^1$H–$^1$H correlation (COSY) NMR spectra of **5′** were performed on the Varian Inova 600 spectrometer using CD$_3$CN as the solvent. The $^{13}$C-NMR spectrum of **7** was

performed on the Varian ui500 spectrometer using $CD_3CN$ as the solvent. The $^1H$, HMQC and HMBC NMR spectra of **5′** were performed on the 800 MHz Agilent (Varian) VNMRS spectrometer using $CD_3CN$ as the solvent. The $^1H$, HMQC and HMBC NMR spectra of **5** were performed on the Varian Inova 600 spectrometer using $CDCl_3$ as the solvent. The $^1H$-NMR spectra of **3** and **4** were performed on the Varian Inova 600 spectrometer using $CD_3CN$ as the solvent, and the $^1H$-NMR spectra of **1** and **9** were performed on the Varian Inova 600 spectrometer using DMSO-d6 as the solvent. High-resolution mass spectrometry data were obtained on a coupled Agilent 6520 Accurate-Mass Q-TOF ESI mass spectrometer in positive ion mode, and analysed using Mass Hunter Qualitative Analysis software (Agilent).

NMR data of **3** were as follows: $^1H$-NMR (600 MHz, $CD_3CN$, in formate form): δ 7.05 (d, $J = 8.4$, 1H), 6.95 (d, $J = 8.4$, 1H), 6.79 (s, 1H), 6.75 (s, 1H), 6.00 (d, 1H, 1.2), 5.98 (d, 1H, 1.2), 4.73 (d, 1H, 16.8), 4.60 (d, 1H, 16.2), 4.56 (m, 1H), 3.85 (s, 3H), 3.83 (s, 3H), 3.68 (m, 1H), 3.46 (m, 1H), 3.42 (m, 1H), 3.08 (m, 1H), 3.21 (m, 2H) and 3.16 (m, 3H). This agrees with previous reported NMR data of **3** (refs 48,71). HRMS ($m/z$): $[M]^+$ calcd. for $C_{21}H_{24}Cl_2NO_4^+$, 354.1705; found, 354.1706.

NMR data of **4** were as follows: $^1H$-NMR (600 MHz, $CD_3CN$, in formate form): δ 7.03 (d, $J = 8.4$, 1H), 6.91 (d, $J = 8.4$, 1H), 6.28 (s, 1H), 5.85 (s, 2H), 5.22 (m, 1H), 4.67 (d, 1H, 16.2), 4.78 (d, 1H, 16.2), 3.85 (s, 3H), 3.83 (s, 3H), 3.60 (m, 1H), 3.31 (m, 1H), 3.49 (dd, 1H, 5.4, 18), 2.86 (dd, 1H, 11.4, 18), 3.19 (m, 1H), 3.06 (m, 1H) and 3.14 (s, 3H). This agrees with previous reported NMR data of **4** (refs 48,71). HRMS ($m/z$): $[M]^+$ calcd. for $C_{21}H_{24}Cl_2NO_5^+$, 370.1654; found, 370.1655.

NMR data of **5′** were as follows: $^1H$-NMR (800 MHz, $CD_3CN$): δ 10.22 (s br, 1H), 7.14 (d, $J = 8.8$, 1H), 6.93 (d, $J = 8.8$, 1H), 6.23 (s, 1H), 5.88 (dd, $J = 0.8$, 4, 2H), 3.90 (s, 3H), 3.87 (s, 3H), 3.83 (m, 1H), 3.22 (m, $J = 2.4$, 4.8, 10.4, 1H), 3.20 (dd, $J = 11.2$, 14.4, 1H), 2.90 (dd, $J = 9.6$, 13.6, 1H), 2.76 (m, $J = 4$, 6.4, 1H), 2.62 (d, $J = 11.2$, 1H), 2.34 (d, $J = 16$, 1H), 2.19 (s, 3H). HRMS ($m/z$): $[M + H]^+$ calcd. for $C_{21}H_{24}NO_6^+$, 386.1604; found, 386.1597.

NMR data of **5** were as follows: $^1H$-NMR (600 MHz, $CDCl_3$): δ 7.20 (d, $J = 8.4$, 1H), 6.95 (d, $J = 8.4$, 1H), 6.05 (s, 1H), 5.86 (d, $J = 4.8$, 2H), 5.28 (d, $J = 4.8$, 1H), 4.93 (s, 2H), 4.88 (d, 1H, $J = 5.4$), 3.874 (s, 3H), 3.865 (s, 3H), 3.38 (s, 3H), 3.34 (m, 2H), 3.15 (m, 1H) and 2.92 (dd, 1H, $J = 4.8$, 24). This agrees with previous reported NMR data of **5** (ref. 47). HRMS ($m/z$): $[M]^+$ calcd. for $C_{21}H_{24}NO_6^+$, 386.1604; found, 386.1598.

NMR data of **6** were as follows: $^1H$-NMR (600 MHz, $CD_3CN$, in formate form): δ 7.08 (d, $J = 8.4$, 1H), 7.03 (d, $J = 8.4$, 1H), 6.21 (s, 1H), 6.13 (d, $J = 6.6$, 1H), 5.84 (d, $J = 6.6$, 2H), 5.48 (d, $J = 6.6$, 1H), 4.86 (s, 2H), 3.87 (s, 3H), 3.86 (s, 3H), 3.53 (m, 1H), 3.26 (m, 1H), 3.21 (s, 3H), 3.05 (m, 2H) and 2.03 (s, 3H). This agrees with previous reported NMR data of **6** (ref. 47). HRMS ($m/z$): $[M]^+$ calcd. for $C_{23}H_{26}NO_7^+$, 428.1709; found, 428.1708.

NMR data of **7** were as follows: $^1H$-NMR (600 MHz, $CD_3CN$): δ 10.06 (s, 1H), 7.09 (s, 2H), 6.36 (d, $J = 5.4$ Hz, 1H), 6.17 (s, 1H), 5.85 (d, $J = 1.2$, 1H), 5.88 (d, $J = 1.2$, 1H), 4.14 (d, $J = 5.4$, 1H), 3.85 (s, 3H), 3.82 (s, 3H), 2.90 (td, $J = 4.2$, 10.2, 1H), 2.37 (s, 3H), 2.32 (dt, $J = 3.6$, 15.6, 1H), 2.21 (dt, $J = 3.6$, 10.8, 1H), 2.05 (ddd, $J = 4.2$, 10.8, 1H), 1.99 (s, 3H). $^{13}C$-NMR (500 MHz, $CD_3CN$): δ 191.74, 170.73, 153.22, 150.98, 148.03, 138.77, 133.67, 133.11, 131.28, 131.04, 125.42, 116.86, 116.15, 102.04, 101.11, 76.09, 62.27, 62.22, 56.49, 51.45, 45.76, 28.95 and 21.30. HRMS ($m/z$): $[M + H]^+$ calcd. for $C_{23}H_{26}NO_8^+$, 444.1658; found, 444.1655.

NMR data of **9** were as follows: $^1H$-NMR (600 MHz, DMSO-d6, in formate form): δ 7.22 (d, $J = 8.4$, 1H), 6.31 (s, 1H), 5.99 (dd, $J = 1.2$, 9.6, 2H), 5.95 (dd, $J = 0.6$, 8.4, 1H), 5.67 (dd, $J = 0.6$, 9.6, 1H), 4.31 (d, $J = 4.2$, 1H), 3.87 (s, 3H), 3.79 (s, 3H), 2.43 (s, 3H), 2.43 (m, 1H), 2.36 (m, 1H), 2.21 (m, 1H), 1.80 (m, 1H). This agrees with previous reported NMR data of **9** (ref. 72,73). HRMS ($m/z$): $[M + H]^+$ calcd. for $C_{21}H_{22}NO_7^+$, 400.1396; found, 400.1394.

NMR data of **1** were as follows: $^1H$-NMR (600 MHz, DMSO-d6, in formate form): δ 7.27 (1H), 6.52 (1H), 6.17 (1H), 6.00 (2H), 5.54 (1H), 4.23 (1H), 3.95 (3H), 3.86 (3H), 3.81 (3H), 2.43 (3H), 2.55 (1H), 2.43 (1H), 2.29 (1H) and 1.88 (1H). All the signals of **1** appear to be broad peaks in DMSO-d6, possibly due to chemical exchange. The chemical shifts and integration of peaks agree with previous reported NMR data of **1** (refs 72,73). HRMS ($m/z$): $[M + H]^+$ calcd. for $C_{22}H_{24}NO_7^+$, 414.1553; found, 414.1554.

**O-Methyltransferase expression and purification.** pCS3306 and pCS3307 were transformed individually and in combination into *E. coli* strain BL21(DE3) for PsMT2 or PsMT3 homodimer expression and PsMT2 and PsMT3 co-expression. The cells were cultured at 37 °C and 250 r.p.m. in 500 ml of lysogeny broth medium supplemented with 50 μg ml$^{-1}$ kanamycin and/or 100 μg ml$^{-1}$ ampicillin to a final OD600 between 0.4 and 0.6. The culture was then incubated on ice for 10 min before addition of 0.4 mM isopropylthio-β-D-galactoside to induce protein expression. The cells were further cultured at 16 °C for 12–16 h. The cells were harvested by centrifugation (2,164g, 15 min, 4 °C), resuspended in 25 ml lysis buffer (20 mM Tris-HCl, pH 7.9, 0.5 M NaCl and 10 mM imidazole), and lysed by sonication on ice. Cellular debris was removed by centrifugation (27,167g, 20 min, 4 °C), and appropriate affinity agarose resins were then added to the supernatant (2 ml l$^{-1}$ of culture, Ni-NTA agarose resin for PsMT2, and T7 tag antibody agarose resin for PsMT3). The suspension was swirled at 4 °C for 2 h and loaded onto a gravity column.

For the purification of PsMT2, the PsMT2-bound resin was washed with 50 column volume of 10 mM imidazole in buffer A (50 mM Tris-HCl, pH 7.9, 2 mM EDTA and 2 mM dithiothreitol (DTT)), followed by 20 column volume of 20 mM imidazole in buffer A. PsMT2 was then eluted with 250 mM imidazole in buffer A. For the purification of PsMT3, the PsMT3-bound resin was washed with 10 column volume of binding buffer (4.29 mM $Na_2HPO_4$, 1.47 mM $KH_2PO_4$, 2.7 mM KCl, 137 mM NaCl, 0.1% Tween-20, 0002% sodium azide and pH 7.3). The T7-tagged protein was eluted with T7 Tag Elute Buffer (100 mM citric acid, pH 2.2), and each 1 ml of elute was neutralized with 150 μl of the T7 Tag Neutralization Buffer (2 M Tris base, pH 10.4). For the purification of PsMT2/PsMT3 heterodimer, the supernatant of the cell lysate was first incubated with Ni-NTA agarose resin, and the Ni bound protein was isolated following the same procedure as that of PsMT2. The elution from the Ni-NTA agarose resin was concentrated and exchanged into 25 ml binding buffer, and incubated with T7 tag antibody agarose resin. The PsMT2/PsMT3 heterodimer was then purified following the same procedure for the purification of PsMT3. Purified enzymes were concentrated and exchanged into storage buffer (50 mM sodium phosphate, pH 7.5, 100 mM NaCl, 10% glycerol and 2 mM DTT) with the centriprep filters (Amicon), aliquoted and flash frozen using acetone/dry ice. Protein concentration was determined with the Bradford assay[74] using bovine serum albumin as a standard, and purified PsMT2 and PsMT3 heterodimer was analysed on SDS–PAGE gel.

The PsMT2/Ps6OMT heterodimer was purified from *E. coli* co-transformed with pCS3306 and pCS3534 following a similar procedure as described for the PsMT2/PsMT3 purification process.

**Western blot analysis of O-methyltransferases.** Prestained protein standard and 0.5–2 μg (0.5 μg for Anti-T7 western blot, and 2 μg for Anti-His western blot) of the purified PsMT2 homodimer, PsMT3 homodimer, PsMT2/PsMT3 heterodimer and PsMT3/Ps6OMT heterodimer were loaded on a NuPage Novex 4–12% Bis-Tris protein gel and run with NuPAGE MOPS SDS running buffer (Life Technologies) at 175 V for 45 min. The protein material was then semi-dry transferred to a nitrocellulose membrane or polyvinylidene difluoride membrane at 15 V for 15 min with transfer buffer (Life Technologies). The membrane was blocked with 5% bovine serum albumin in tris buffered saline and Tween-20 (TBST) buffer (50 mM Trizma base, 150 mM NaCl, 0.05% Tween-20 and pH 8.0) for 1 h, and then probed overnight at 4 °C with Anti-6X His tag antibody (ab1269 from Abcam PLC, 1:5000) or Anti-T7 tag antibody (ab21477 from Abcam PLC, 1:5000) conjugated with HRP in TBST buffer for 12–16 h. Chemiluminescence was induced by SuperSignal West Pico substrate (Pierce) and images were acquired by a G:Box Chemi XT4 imaging system (Syngene).

**In vitro reconstitution of O-methyltransferases.** The in vitro methyltransferase reactions were performed as previously described[50] at 50 μl scale containing 100 mM Gly-NaOH buffer (pH 9.0) in the presence of 25 mM sodium ascorbate, 2 mM DTT, 10% glycerol, 300 μM S-Adenosyl methionine, 10 μM purified **9** and 50 μg purified enzyme or no enzyme as a control. The reactions were performed overnight and quenched with 150 μl of 99.9% MeOH/0.1% formic acid. The quenched reaction mixture was analysed on an Agilent 6420 triple quad LC–MS (Agilent EclipsePlus C18, 2.1 × 50 mm, 1.8 μm) using positive ionization. Noscapine and narcotoline were detected with MRM using the highest characteristic precursor ion/product ion transition (Supplementary Table 8).

**Size-exclusion chromatography analysis.** The PsMT2/PsMT3 and PsMT2/Ps6OMT enzyme complex as a dimer were determined by SEC on a normal-phase HPLC (Agilent) using the BioSep-SEC-s4000 column. The column was equilibrated and eluted with 300 mM NaCl in 100 mM sodium phosphate buffer, pH 7.5, at a flow rate of 1 ml min$^{-1}$, and calibrated with the following protein standards: sweet potato β-amylase (200 kDa); yeast alcohol dehydrogenase (150 kDa); bovine serum albumin (66 kDa); bovine erythrocytes carbonic anhydrase (29 kDa) and horse heart cytochrome c (12.4 kDa). Protein elution was detected by absorbance at 280 nm.

**Data availability.** The authors declare that the data supporting the findings of this study are available within the article and its Supplementary Information files.

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

## Acknowledgements

We thank the Stanford Chemistry NMR Facility and S. Lynch for instrument access and training; Dr. C. Liu of the Stanford Magnetic Resonance Laboratory for assistance with 800 MHz NMR experiments, the SMRL 800 MHz NMR spectrometer was supported in part by NIH Shared Instrumentation Grant (1 S10 RR025612-01A1); Professor C. Khosla for access to materials and instruments; Professor E.S. Sattely for access to the Agilent 6520 Accurate-Mass Q-TOF ESI mass spectrometer; X. Xiao, X. Li for assistance and discussion in protein expression, purification and characterization; R.J. Li for assistance in obtaining the high-resolution mass spectrometry data; D. Endy, S. Galanie, S. Li, M. McKeague, P. McLean, C. Schmidt, K. Thodey and I. Trenchard for valuable feedback in the preparation of the manuscript; Y.-H. Wang for assistance and discussion in data analysis. This work was supported by the National Institutes of Health (grant to C.D.S., AT007886) and Novartis Institutes for Biomedical Research (grant to C.D.S., IC2013-1373).

## Author contributions

Y.L. and C.D.S. conceived of the project, designed the experiments, analysed the results and wrote the manuscript. Y.L. performed the experiments.

## Additional information

**Competing financial interests:** S.C.D. and L.Y. report a patent application published on May 26, 2016 'Noscapinoid-Producing Microbes and Methods of Making and Using the Same'. Application no. WO 2016/081371.

