## [Peer review file · Nature Communications]

Reviewers' comments:

Reviewer #1 (Remarks to the Author):

Li and Smolke report the reconstitution of the norlaudanosoline to noscapine pathway in yeast. The manuscript also offers the first description of narcotoline 4'-O methyltransferase, which the authors find to function as a heterodimer. Using promiscuous enzymes, the authors are able to produce a novel pathway intermediate. The authors have thoroughly characterized the non-commercially available intermediates using MS and NMR. The authors claim this pathway to be one of the most complex plant pathway reconstitutions; however, as I elaborate below, this reviewer does not agree with this claim.

The novelty of this work is the first microbial production of the noscapine from norlaudanosoline. However, the scientific impact of this work is minimal. The pathway reconstitution was a tour de force, but relatively straightforward given that all enzymes were previously known. No novel engineering strategies or techniques were used to achieve noscapine production. The promoter, temperature, and N-terminus optimization, especially of CYPs, is standard in the field and is done whenever a new pathway is engineered. While the authors converted norlaudanosoline to noscapine, the conversion of norlaudanosoline to reticuline and reticuline to canadine has already been achieved (by the same group). Thus, only the conversion of canadine to noscapine is truly novel. The only scientific contribution is the discovery of a need for co-expression of heterodimer components.

Overall, the manuscript would be of interest to the alkaloid biosynthesis field because it is the first production of noscapine. However, I think it would be of limited interest to the general Synthetic Biology/ Chemical Biology Community, given that the manuscript offers very little scientific novelty, and no new strategies or techniques were used to achieve the production of the chemical. Therefore, the manuscript's influence on the field will be limited. A number of scientifically strong alkaloid bioproduction papers have been published in the past year and this one is not at the same level.

Major comments

The authors should comment in the discussion on why it was not attempted to produce noscapine de novo. Although norlaudanosoline has not been produced from glucose de novo in yeast, (S)-reticuline has, through the intermediate norcoclaurine. To produce noscapine from glucose, the authors could either use previous strategies from yeast to produce (S)-reticuline via a norcoclaurine intermediate by adding N-methylcoclaurine hydroxylase (NMCH) (e.g. DeLoache et al. [Dueber], Nat Chem Biol, 2015; Galanie et al. [Smolke] Science 2015; Trenchard et al. [Smolke] Metab Eng, 2015) or via a norlaudanosoline intermediate as has been done in E.coli by expressing monoamine oxidase (MAO) to convert dopamine to DHPAA (Nakagawa, Nat Commun, 2011; Nakagawa, Nat Commun, 2014). The condensation of dopamine and DHPAA (instead of HPAA) would produce norlaudanosoline and remove the necessity of NMCH in the production of reticuline. If the de novo biosynthesis of noscapine was attempted, the authors should discuss the yields obtained, the

limitations in the pathway and propose key enzyme that need to be improved to obtain significant yields of noscapine de novo. Because scientists in the field will know that de novo production of noscapine could have been attempted, not bringing it up in the discussion is concerning.

This reviewer disagrees with the claim that this work is one of the most "complex" plant pathways constructed in yeast. The pathway in this manuscript is not even the longest (although this reviewer does not know what value that would add), according to the same authors' previous work: Galanie et al. [Smolke] Science 2015: "..., including expression of 21 heterologous enzymes from plants, mammals, bacteria, and yeast, overexpression of two native yeast enzymes, and deletion of one native yeast gene". This reviewer believes that the most complex plant pathway engineered in yeast to date is Brown, et al [O'Connor] PNAS 2015, which included not only expression of 16 heterologous enzymes, but three gene deletions and five chromosomal overexpressions to achieve the de novo synthesis of strictosidine: "The best of our engineered strains (strain 4 containing G8H plasmid) harbored three gene deletions (Δ ERG20, Δ ATF1, Δ OYE2), 15 plant-derived genes (AgGPPS2,GES,G8H,GOR,ISY,IO,7-DLGT,7-DLH,LAMT,SLS,TDC,STR,CYB5,CP R,CYPADH), one animal-derived gene (avian mutant FPPS:mFPSN144W), and five additional copies of yeast genes (tHMGR,MAF1,IDI1,SAM2,ZWF1) to allow de novo production of strictosidine." Given that the definition of "complex pathway" is subjective, I would advise the authors to remove this phrasing from the manuscript.

To calculate the conversion rate, the authors use grams rather than moles, which overestimates their conversion rate given that hydroxyl and methyl groups are continuously added to the molecule. For example, 36 mg/L of (S)-1-hydroxy-N-methylcanadine, 4, from 85 mg/L fed (S)-canadine, 2, is calculated as a 42% conversion, but using moles, which takes into account the amount of 2 converted, it is only 39%. g/g conversions are acceptable when the compound is produced de novo, the idea being that that one knows how much glucose was introduced to the system to produce the product. However, when starting from an advanced intermediate, it is misleading. Even more so when starting from two different intermediates (norlaudanosoline and canadine).

Since racemic substrates were fed to the yeast, was there any mix of enantiomers produced at any step of the pathway? Was the "wrong" enantiomer of the starting material still present in the culture in all cases? When stating % conversions from a single enantiomer, the assumption of equal amounts of each enantiomer should be stated.

It would be useful to include P values for some of the data points. In some cases, it is claimed that there is an increase in production, however from the figure it seems statistically insignificant (e.g., Page 6: ~8mg/L conversion of 2 to 4 versus ~10mg/L and a Page 14:~50% increase in noscapine production between strains CSYN16 and CSYN17_2).

The writing in the section analyzing the PsMT2/PsMT3 heterodimer is unclear and does not fully support the conclusions. In multiple figures, this step is shown as "PsMT2 + PsMT3 or

Ps6OMT", however in the text it is often referred to as just PsMT2/PsMT3. Further, when referencing figures after talking about the PsMT3 dimer, the strain in the figure is with the Ps6OMT dimer. Specifically: "...the 4'-O-methylation is likely to be naturally catalyzed by the PsMT2/PsMT3 heterodimer (Supplementary Fig. 4a)." This figure represents the transition as "PsMT2 + PsMT3 or Ps6OMT", however only data from the Ps6OMT strain is shown, according to the supplemental table, contrary to the statement in the manuscript and figure heading. It would also strengthen the claims if data was presented for both heterodimers. Is there data supporting that PsMT3 is better or more efficient than Ps6OMT? From figure 2i, it appears that the Ps6OMT dimer leads to higher production of noscapine. Were Western blots completed to show the PsMT2/Ps6OMT heterodimer?

The fact that 4'-O-desmethyl-3-O-acetylppapaveroxine, 7, spontaneously cyclizes to form narcotolinehemiacetal, 8, should be addressed earlier than the removal of the carboxylesterase PsCXE1 from the narcotoline, 9,-producing strain, as 8 can be seen as early as the 7-producing strain. Also, the presence of 8 is not merely "implied" but shown in the various data. Further, in supplemental figure 3a-ii, there appears to be a peak representing production of 8, however the acetyltransferase PsAT wasn't expressed in this strain. Is it possible that peak is not solely 8? In general, since the LC method is so short and it appears that compounds elute from the column at similar or overlapping times, what did the LC-MS scan data look like? Were the samples relatively clean or dirty? Could ion suppression be an issue?

The article is written in very technically. The authors would benefit from simplifying the writing and broadening the discussion to make it more appealing to the wider audience of Nature Communications. For example, the authors could discuss what would need to be addressed to produce noscapine and related compounds de novo from an inexpensive simple sugar instead of an expensive intermediate, including such options as co-culturing with norlaudanoline producing microbes. How far is this strain from competing with noscapine levels in plants, since the large drawback to plant extraction is the low native yield?

The authors heavily reference their own work, however there are number of papers (some of them more contemporary than the citations chosen) that should also be included when supporting some of the statements. The authors would be advised to reference previous work more thoroughly.

Minor comments

There are a handful of typos in the methods section. Also, some of the methods can go into the supplementary information (e.g. plasmid and yeast construction, NMR shifts).

It appears that all titers reported were of extracellular metabolite levels. The authors state that "all major pathway intermediates are efficiently transported out of yeast cells to the medium." However, in the referenced figure, Supplemental figure 6, it appears that while most of the metabolites are transported out of the cell, the efficiency is poor in some cases (e.g. 7, 6),

leaving a significant amount of metabolite inside the cell. The authors would benefit from highlighting this point in the discussion and include that loss of intermediates to the media (e.g. reticuline, scoulerine) could contribute to poor flux through the pathway. State in the results section that the numbers provided are for extracellular noscapine

In the title, the authors refer to noscapine as an anticancer agent but elsewhere as a potential anticancer agent. Is noscapine itself currently used as. or is it an advanced intermediate for, a current therapeutic? It could strengthen the manuscript if the authors elaborated on the importance of both noscapine and the family of molecules in general (this is alluded to when stating that "few studies have been conducted on secoberberine-based drug development" due to inaccessible intermediates).

Multiple figure references in the text referred to parts of figures which had none (e.g. Fig. 1b, 4a). Further, some figures are referenced out of numeric order (e.g. Figure 5 is referred to in text before Figure 4).

At the end of the paragraph about expressing CYP82X1, it is mentioned that "The highest activity...was observed when the enzyme was expressed from relatively weak promoters (ADH1, CYC1)..." While included in the supplemental figure, the ADH1 bar was not included in the main figure in the manuscript, although the HXT bar was. Further, since HXT (and ADH1 according to the supplemental table) was chosen for expression, HXT should probably be included in the parenthetical, as production in all three cases was relatively equal. On a related note, in the paragraph referring to expressing PsCXE1, it is stated that "we engineered the 7-producing strain to express PsCXE1 downstream of a strong promoter (PYK1, GPD) and observed..." As both promoters are used in various strains as well, how did production levels compare using the different promoters?

Some of the strains seem incorrect in the supplemental table. Some examples: the table shows the same strain used for figures 2G-ii, 2H-ii, and 2I-ii, however these seem to be 3 unique strains according to the figure legend; same for the strain listed for figures 2H-v and 2I-iv.

All final plasmids should be listed in Supplemental Table 2, not just plasmids used to create other plasmids. Further, "For plasmid-based DNA assembly, certain cassettes were PCR amplified and incorporated into pYES1L as described previously to yield..." should have a citation as this wasn't explained previously in this work. A number of plasmids appear to not have explanations for how they were made (e.g. pCS3149, pCS3173-3175).

When converting to the norlaudanosoline fed strain, the enzyme norcoclaurine 6-O-methyltransferase (Ps6OMT) was added. The reviewer is curious if this has an effect on the investigated O-methylation step later in the pathway as this enzyme was shown to also create the heterodimer and catalyze the reaction to produce noscapine. It would be interesting to see what effect removing PsMT3 and lessening the heterologous burden on the cell would have on noscapine titers.

The authors should provide the DNA sequences of the enzymes as they were optimized for expression either in the supplementary information or include their Genbank codes in the paper or methods section.

In the methods section, under Culture Conditions for Metabolite Production, it appears that productions were done using galactose as a carbon source. As no enzymes were under galactose-inducible promoters, is this correct? This should be stated in the text.

While mentioned in the methods section, it should be mentioned in the text that some standards were unavailable and thus structurally similar standards were used to quantify titers. Further, were internal standards included in sample analysis?

Figure 1. In Figure 1, it appears that Ps6OMT (in the reaction converting scoulerine to tetrahydrocolumbamine) is mislabeled as TfS9OMT. It would also be helpful, to understand the full system from the figure, if AtATR1 was included somewhere in the figure. There also appears to be a typo in the compound name of 4.

Figure 2. In Figure 2, include traces for standards when available.

Figure 3. In the caption of Figure 3c, it's stated "through varying promoter and copy number"; however, only the promoters were varied in the figure. In Figure 3d, 2 should be racemic; this should be reflected in the structure and the y-axis. Also, the strains appear to be labeled incorrectly in the figure and caption (CSYN10 and 11 instead of 9 and 10 as listed in the supplemental table).

Reviewer #2 (Remarks to the Author):

The authors describe the construction of a strain of yeast capable of producing the alkaloid noscapine. This accomplishment is achieved through the step-wise reconstruction of an already elucidated plant pathway/gene cluster. The result is a nice accomplishment for the field of secondary metabolic production. The paper describes a unique chemistry pathway and provides proper citations for the field. This paper will be a nice contribution to the journal.

Reviewer #3 (Remarks to the Author):

This manuscript reports the assembly of the complete pathway for the biosynthesis of the medicinal alkaloid noscapine in yeast (*Saccharomyces cerevisiae*). This is an exciting development that follows on from the work from the Graham lab at the University of York (UK) reporting the identification and partial characterization of a ten gene metabolic cluster for the noscapine pathway in poppy. Increasingly more complex plant natural product pathways have been recently assembled in yeast, including those for opioids from poppy from the Smolke lab and also the strictosidine pathway from Madagascan periwinkle from the O'Connor lab. Despite the work from the Graham lab and others working on the

noscapine pathway (e.g. the Facchini lab) the full pathway has until now not been fully characterized and questions about certain pathway steps have remained outstanding.

The work presented here is elegant and pioneering. A series of steps was taken to engineer the ability to synthesise noscapine from canadine. Various limitations were overcome, including optimisation of the limiting step catalysed by CYP82Y1. This was achieved by optimising the N-terminal tag used for expression of CYP82Y1, the promoters and the growth temperatures. This resulted in a strain optimized for the production of compound 4, which was then used in investigations of downstream steps that provide important support for the sequence of events previously reported from in vitro studies involving CYP82X2, PsAT1 and CYP82X1. Critically, the optimized yeast strain plus these engineered and optimized subsequent steps was able to generate sufficient secoberberine intermediates for full structural characterization by NMR (which had not previously been achieved due to lack of compound availability). A key finding is that the 4'-O-methylation step appears to be catalysed by an O-methyl heterodimer. This is a very important discovery. Careful analysis is carried out in yeast to demonstrate that the two methyl transferases are indeed physically associated with each other and that neither alone is sufficient to catalyse the 4'-O-methylation step. The authors then go on to demonstrate that they are able to synthesise other protoberberine compounds by exploiting 'off-pathway' enzyme activities, and to also engineer into yeast the ability to produce noscapine from the early BIA backbone norlaudanosoline.

There is no analysis of enzyme kinetics included in the work. However monitoring the effectiveness of the different pathway steps in terms of mgs product/L as the authors have done is in essence a more direct way of evaluating performance in the yeast system. The approach that they have taken is pragmatic - identifying bottlenecks and overcoming them by making and testing different design formats. The demonstration that the two methyl transferases work in yeast as a heterodimer is particularly intriguing. The applicants go to some lengths to explain in the introduction why validation of enzyme function in planta can give misleading results. However it is also the case that expression of plant genes and enzymes in heterologous hosts such as yeast may also lead to artefacts. While I do not regard it as essential for the publication of this work, investigation of whether such a heterodimer exists in poppy is an obvious priority for future work.

One minor point - Claims of priority ('This is the first...') should be avoided.

Response to Reviewers' Comments

We would like to thank the reviewers for their constructive comments and thoughtful suggestions for improving our manuscript. The comments were very useful in helping us revise the manuscript. Below we address each comment made by the reviewers.

Reviewer #1 (Remarks to the Author):

Li and Smolke report the reconstitution of the norlaudanoline to noscapine pathway in yeast. The manuscript also offers the first description of narcotoline 4'-O methyltransferase, which the authors find to function as a heterodimer. Using promiscuous enzymes, the authors are able to produce a novel pathway intermediate. The authors have thoroughly characterized the non-commercially available intermediates using MS and NMR. The authors claim this pathway to be one of the most complex plant pathway reconstitutions; however, as I elaborate below, this reviewer does not agree with this claim.

The novelty of this work is the first microbial production of the noscapine from norlaudanoline. However, the scientific impact of this work is minimal. The pathway reconstitution was a tour de force, but relatively straightforward given that all enzymes were previously known. No novel engineering strategies or techniques were used to achieve noscapine production. The promoter, temperature, and N-terminus optimization, especially of CYPs, is standard in the field and is done whenever a new pathway is engineered. While the authors converted norlaudanoline to noscapine, the conversion of norlaudanoline to reticuline and reticuline to canadine has already been achieved (by the same group). Thus, only the conversion of canadine to noscapine is truly novel. The only scientific contribution is the discovery of a need for co-expression of heterodimer components.

Overall, the manuscript would be of interest to the alkaloid biosynthesis field because it is the first production of noscapine. However, I think it would be of limited interest to the general Synthetic Biology/ Chemical Biology Community, given that the manuscript offers very little scientific novelty, and no new strategies or techniques were used to achieve the production of the chemical. Therefore, the manuscript's influence on the field will be limited. A number of scientifically strong alkaloid bioproduction papers have been published in the past year and this one is not at the same level.

We thank the reviewer for highlighting this first demonstration of metabolic engineering for the microbial biosynthesis of this group of compounds, and pointing out that the discovery of the narcotoline 4'-O-methyltransferase is a novel and exciting finding in the field of plant biosynthesis very broadly.

In addition, we are excited about being able to provide the structural verification of the key intermediates (compounds 5' and 7), show the reaction matrix among the final three steps in yeast, and provide the sourcing of many major pathway intermediates, especially secoberberines, through the step-wise reconstitution work. We understand that most of the enzymes encoded in the noscapine gene cluster are characterized or understood from the pioneering *in planta* and *in vitro* characterization studies. However, there remained unclear steps, and most of the pathway intermediates were inaccessible. We used the reconstitution of the noscapine gene cluster as an

initial showcase to demonstrate that such complex plant gene cluster reconstitution work in yeast can complement the *in planta* and *in vitro* characterizations, and can advance the characterization of related enzymes and molecules. Because of that, we believe this work will be of interest to not only the alkaloid biosynthesis field, but also researchers interested in studying plant specialized metabolism and biosynthesis of related valuable molecules. Also, the discovery of the narcotoline 4'-O-methyltransferase as a heterodimer will be of interest to the broader plant natural product biosynthesis field.

We also thank the reviewer for the suggestions for improvement, and have made corresponding edits as detailed below.

Major comments:

The authors should comment in the discussion on why it was not attempted to produce noscapine de novo. Although norlaudanosoline has not been produced from glucose de novo in yeast, (S)-reticuline has, through the intermediate norcoclaurine. To produce noscapine from glucose, the authors could either use previous strategies from yeast to produce (S)-reticuline via a norcoclaurine intermediate by adding N-methylcoclaurine hydroxylase (NMCH) (e.g. DeLoache et al. [Dueber], Nat Chem Biol, 2015; Galanie et al. [Smolke] Science 2015; Trenchard et al. [Smolke] Metab Eng, 2015) or via a norlaudanosoline intermediate as has been done in E. coli by expressing monoamine oxidase (MAO) to convert dopamine to DHPAA (Nakagawa, Nat Commun, 2011; Nakagawa, Nat Commun, 2014). The condensation of dopamine and DHPAA (instead of HPAA) would produce norlaudanosoline and remove the necessity of NMCH in the production of reticuline. If the de novo biosynthesis of noscapine was attempted, the authors should discuss the yields obtained, the limitations in the pathway and propose key enzyme that need to be improved to obtain significant yields of noscapine de novo. Because scientists in the field will know that de novo production of noscapine could have been attempted, not bringing it up in the discussion is concerning.

We thank the reviewer for pointing out the possibility of the *de novo* production of noscapine, and completely agree that this is an important part of the future work in this space. According to previous work from our group and others, the *de novo* production of reticuline is limited (<0.1 mg/L); and to achieve the *de novo* biosynthesis of noscapine, no less than 11 additional enzymatic steps are required to be introduced into the reticuline producing strain, which include an additional four plant cytochrome P450s. Therefore, additional optimization efforts will be needed to achieve production of noscapine *de novo*. We have added a paragraph in the discussion section focused on the potential of *de novo* production of noscapine based on this and other work in the plant alkaloid space.

This reviewer disagrees with the claim that this work is one of the most "complex" plant pathways constructed in yeast. The pathway in this manuscript is not even the longest (although this reviewer does not know what value that would add), according to the same authors' previous work: Galanie et al. [Smolke] Science 2015: "..., including expression of 21 heterologous enzymes from plants, mammals, bacteria, and yeast, overexpression of two native yeast enzymes, and deletion of one native yeast gene". This reviewer believes that the most complex plant pathway engineered in yeast to date is Brown, et al [O'Connor] PNAS

*2015, which included not only expression of 16 heterologous enzymes, but three gene deletions and five chromosomal overexpressions to achieve the de novo synthesis of strictosidine: "The best of our engineered strains (strain 4 containing G8H plasmid) harbored three gene deletions (Δ ERG20, Δ ATF1, Δ OYE2), 15 plant-derived genes (AgGPPS2,GES,G8H,GOR,ISY,IO,7-DLGT,7-DLH,LAMT,SLS,TDC,STR,CYB5,CPR,CYPADH), one animal-derived gene (avian mutant FPPS:mFPSN144W), and five additional copies of yeast genes (*t*HMGR,MAF1,IDII,SAM2,ZWF1) to allow de novo production of strictosidine." Given that the definition of "complex pathway" is subjective, I would advise the authors to remove this phrasing from the manuscript.*

We thank the reviewer for pointing this out. We originally did not take into account the biosynthesis of the primary metabolites (i.e., tyrosine, isopentenyl pyrophosphate and dimethylallyl pyrophosphate, respectively, for the synthesis of hydrocodone and strictosidine) from sugar, and in that case the three pathways are each composed of 13-14 heterologous enzyme-catalyzed steps. Based on the similarity in the number of heterologous steps we included the description as "one of the most complex plant pathways" in contrast to "one of the most engineered plant natural product producing yeast strains". However, we realize that the definition of "complex pathway" is very subjective, and we have revised the text to remove such language from the manuscript.

*To calculate the conversion rate, the authors use grams rather than moles, which overestimates their conversion rate given that hydroxyl and methyl groups are continuously added to the molecule. For example, 36 mg/L of (*S*)-1-hydroxy-*N*-methylcanadine, 4, from 85 mg/L fed (*S*)-canadine, 2, is calculated as a 42% conversion, but using moles, which takes into account the amount of 2 converted, it is only 39%. g/g conversions are acceptable when the compound is produced de novo, the idea being that that one knows how much glucose was introduced to the system to produce the product. However, when starting from an advanced intermediate, it is misleading. Even more so when starting from two different intermediates (norlaudanosoline and canadine).*

We thank the reviewer for pointing this out and have changed the compound quantification into molar concentration and conversion throughout the revised manuscript.

Since racemic substrates were fed to the yeast, was there any mix of enantiomers produced at any step of the pathway? Was the "wrong" enantiomer of the starting material still present in the culture in all cases? When stating % conversions from a single enantiomer, the assumption of equal amounts of each enantiomer should be stated.

We thank the reviewer for these suggestions to make our data analysis more accurate. Since we don't have access to standards of pure (*S*) or (*R*) forms of the pathway intermediates from canadine to noscapine, we have performed new experiments in which either (*R*)-canadine or (*S*)-canadine was fed to our strains. The new data shows that the enzymes along the pathway from canadine to noscapine can convert both (*S*) and (*R*) forms of canadine into noscapine (Supplementary Fig. 6). However, most pathway enzymes are very specific towards the (*S*) form of the substrates (Supplementary Fig. 6). Thus, when racemic canadine is fed to our engineered yeast strain, compounds synthesized are likely a mixture of enantiomers, with the majority in (*S*)

form; and under these conditions, (*R*)-canadine and other related pathway intermediates in (*R*) form are still present in the culture in all cases. We have included the description and discussion on the new data in the Discussion section. In addition, the manuscript indicates that the canadine fed in most of the experiments is a racemic mixture, which generally indicates that there are equal amounts of left- and right-handed enantiomers. We have revised the manuscript text to clarify the assumption of equal amounts of each enantiomer in the revised manuscript.

It would be useful to include P values for some of the data points. In some cases, it is claimed that there is an increase in production, however from the figure it seems statistically insignificant (e.g., Page 6: ~8mg/L conversion of 2 to 4 versus ~10mg/L and a Page 14:~50% increase in noscapine production between strains CSYN16 and CSYN17_2).

We thank the reviewer for pointing this out and have including the p-values for the data points as suggested by this reviewer. Based on this analysis the increase in noscapine titer of CSYN17_2 relative to CSYN16 is not significant ($p=0.057$), such that increasing the expression level of either CYP82X2 or PsS9OMT does not result in a significant increase in noscapine production. However, enhancing both enzyme (e.g., CYP82X2 and PsS9OMT) expression levels significantly increased noscapine titer, and we have made corresponding changes to the manuscript and figures to make clearer the statements around increases in the production.

The writing in the section analyzing the PsMT2/PsMT3 heterodimer is unclear and does not fully support the conclusions. In multiple figures, this step is shown as "PsMT2 + PsMT3 or Ps6OMT", however in the text it is often referred to as just PsMT2/PsMT3. Further, when referencing figures after talking about the PsMT3 dimer, the strain in the figure is with the Ps6OMT dimer. Specifically: "...the 4'-O-methylation is likely to be naturally catalyzed by the PsMT2/PsMT3 heterodimer (Supplementary Fig. 4a)." This figure represents the transition as "PsMT2 + PsMT3 or Ps6OMT", however only data from the Ps6OMT strain is shown, according to the supplemental table, contrary to the statement in the manuscript and figure heading. It would also strengthen the claims if data was presented for both heterodimers. Is there data supporting that PsMT3 is better or more efficient than Ps6OMT? From figure 2i, it appears that the Ps6OMT dimer leads to higher production of noscapine. Were Western blots completed to show the PsMT2/Ps6OMT heterodimer?

We thank the reviewer for pointing out the unclear description of the PsMT2/PsMT3 heterodimer. According to the pioneering *in planta* study and the fact that PsMT3 is the only remaining O-methyltransferase encoded by the noscapine gene cluster that does not have any function assigned, we proposed that PsMT2 and PsMT3 might both be required for the conversion of narcotoline to noscapine and confirmed this hypothesis via our yeast-based product assay. Since Ps6OMT shares high sequence similarity to PsMT3 (79.5% amino acid sequence identity), we tested the function of Ps6OMT in the synthesis of noscapine from narcotoline as well, and found that Ps6OMT functions similarly to PsMT3. The activities of Ps6OMT and PsMT3 in yeast are comparable (i.e., PsMT3 is not clearly better than Ps6OMT, and vice versa), and that is why we originally showed the Ps6OMT trace in the original Supplementary Figure 4. We realized that the writing and traces in the figures may lead to confusion on our identification of the PsMT2/PsMT3 heterodimer in yeast, and therefore edited the results section and changed the traces in the original Supplementary Figure 4 (now Supplementary Fig. 4b) to those of yeast strains expressing PsMT3. We have also added the

comparison of noscapine titers from strains expressing either PsMT3 or Ps6OMT in Supplementary Figure 4a.

Since PsMT2 and PsMT3 are encoded by the same gene cluster, the regulated expressions of these two genes are likely to be correlated. As Ps6OMT is very well studied in the early part of the benzyloquinoline alkaloid pathway, we do not know if Ps6OMT is natively involved in the synthesis of noscapine from narcotoline in plants, especially due to the fact that different parts of the benzyloquinoline alkaloid biosynthetic pathways in opium poppy follows different spatial and temporal distributions. However, the observed activities of PsMT2, PsMT3, and Ps6OMT in yeast hints at the possibility that functional analysis of the activity of PsMT3 via gene inactivation in poppy might be obscured by the presence of Ps6OMT. As the third reviewer suggests, investigations on whether PsMT2 and PsMT3 form a heterodimer *in planta* and the involvement of Ps6OMT in this reaction (if any) are important aspects of the future work. We have also performed new experiments to show that Ps6OMT and PsMT2 can form a heterodimer and functional narcotoline-4'-O-methyltransferase *in vitro* (this data is now presented in Supplementary Fig. 5).

The fact that 4'-O-desmethyl-3-O-acetylpapaveroxine, 7, spontaneously cyclizes to form narcotolinehemiacetal, 8, should be addressed earlier than the removal of the carboxylesterase PsCXE1 from the narcotoline, 9,-producing strain, as 8 can be seen as early as the 7-producing strain. Also, the presence of 8 is not merely "implied" but shown in the various data. Further, in supplemental figure 3a-ii, there appears to be a peak representing production of 8, however the acetyltransferase PsAT wasn't expressed in this strain. Is it possible that peak is not solely 8? In general, since the LC method is so short and it appears that compounds elute from the column at similar or overlapping times, what did the LC-MS scan data look like? Were the samples relatively clean or dirty? Could ion suppression be an issue?

We thank the reviewer for these suggestions. We did see trace production of a compound of $m/z^+=402$ in the **5**-producing strain expressing CYP82X1, but were unsure if it is the cyclized compound **8** or 4'-O-desmethyl-papaveroxine. The $m/z^+=402$ peak also appears in the **7**- and **8**-producing strains as well. As shown in Supplementary Figure 3c, the MS/MS fragmentations of this peak through different traces are consistent, and the retention time matches through different repeats as well. Since we were not able to purify sufficient amount of this $m/z^+=402$ peak for structural verification via NMR, we cannot determine if the peak is pure compound **8**, and cannot exclude the possibility that the $m/z^+=402$ peaks in the original Supplementary Figure 3a_{ii}, 3a_{iii}, and 3a_{iv} (now Supplementary Fig. 3a_{ii}, 3a_{iv}, and 3a_v) contain multiple components. However, the synthesis of **9** in the absence of PsCXE1 indicates that the peak contains cyclized compound **8** (Supplementary Fig. 3). We also co-expressed PsSDR1 with CYP82X1 in the **5**-producing yeast, and demonstrate that this strain results in trace amount of **9**, which indicates the presence of **8** in this strain as well (Supplementary Fig. 3).

To address the reviewer's comments we have added the observation of trace amount of a compound of $m/z^+=402$ at RT=2.7 min in the description of the **7**-producing yeast. We have modified the text to state the possibility of this peak containing a mixture. We have also added TIC traces of the **8** and **9**-producing yeast cultures, and as shown in Supplementary Fig. 3, there are no strong MS signals that would cause ion suppression.

In addition, the metabolites in the medium were separated on a linear gradient of 10% CH₃CN (v/v in water, 0.1% formic acid) to 50% CH₃CN (v/v in water, 0.1% formic acid) over 5 min with a flow rate of 0.4 mL/min. Although it appears that the retention of some compounds are similar, they do not completely overlap with each other, so they are still quite differentiable through analyzing extracted ion chromatograms. In addition, we also analyzed the peaks by applying multiple reaction monitoring (MRM), which provides additional verification of the compound identity.

The article is written in very technically. The authors would benefit from simplifying the writing and broadening the discussion to make it more appealing to the wider audience of Nature Communications. For example, the authors could discuss what would need to be addressed to produce noscapine and related compounds de novo from an inexpensive simple sugar instead of an expensive intermediate, including such options as co-culturing with norlaudanoline producing microbes. How far is this strain from competing with noscapine levels in plants, since the large drawback to plant extraction is the low native yield?

We thank the reviewer for this suggestion and have modified the language to be more appealing to a wider audience. In particular, we have included additional text in the discussion section on the topics suggested by the reviewer.

The authors heavily reference their own work, however there are number of papers (some of them more contemporary than the citations chosen) that should also be included when supporting some of the statements. The authors would be advised to reference previous work more thoroughly.

We thank the reviewer for this suggestion and have made corresponding changes to the manuscript.

Minor comments

There are a handful of typos in the methods section. Also, some of the methods can go into the supplementary information (e.g. plasmid and yeast construction, NMR shifts).

We thank the reviewer for pointing this out and have gone through the manuscript to identify typos and made corrections in the revised manuscript text. We have also moved some of the methods into the supplementary information as suggested by the reviewer.

It appears that all titers reported were of extracellular metabolite levels. The authors state that "all major pathway intermediates are efficiently transported out of yeast cells to the medium." However, in the referenced figure, Supplemental figure 6, it appears that while most of the metabolites are transported out of the cell, the efficiency is poor in some cases (e.g. 7, 6), leaving a significant amount of metabolite inside the cell. The authors would benefit from highlighting this point in the discussion and include that loss of intermediates to the media (e.g. reticuline, scoulerine) could contribute to poor flux through the pathway. State in the results section that the numbers provided are for extracellular noscapine

We thank the reviewer for this suggestion. We have added a paragraph in the discussion regarding the transport of intermediates along the pathway, how the transport might contribute to

poor flux through the pathway, and how it relates to our measurements of compounds in the media. We have also modified the text in the results section to clarify the metabolites are measured in the medium.

In the title, the authors refer to noscapine as an anticancer agent but elsewhere as a potential anticancer agent. Is noscapine itself currently used as or is it an advanced intermediate for, a current therapeutic? It could strengthen the manuscript if the authors elaborated on the importance of both noscapine and the family of molecules in general (this is alluded to when stating that "few studies have been conducted on secoberberine-based drug development" due too inaccessible intermediates).

We thank the reviewer for this suggestion. Noscapine exhibits anticancer activity, and is currently a potential anticancer agent undergoing preclinical trials, and is also used off-label for cancer treatment. Noscapine is used globally as an antitussive and has been used as an antimalarial drug; it has also shown activity in treating polycystic ovarian syndrome, strokes, and neurodegenerative disorders. We have modified the manuscript text and included additional references in the introduction to provide additional information on noscapine as a potential anticancer agent. We have not found any reports on the bioactivities of secoberberines. Reference 39 is a series of comprehensive reviews summarizing literature on β -phenylethylamines and the isoquinoline alkaloids that include studies on secoberberines; however, no pharmacological data is discussed. We believe this is attributed to the very limited access to these molecules through nature or chemical syntheses.

Multiple figure references in the text referred to parts of figures which had none (e.g. Fig. 1b, 4a). Further, some figures are referenced out of numeric order (e.g. Figure 5 is referred to in text before Figure 4).

We thank the reviewer for pointing this out and have corrected references to figures throughout the manuscript as needed.

At the end of the paragraph about expressing CYP82X1, it is mentioned that "The highest activity...was observed when the enzyme was expressed from relatively weak promoters (ADH1, CYC1)..." While included in the supplemental figure, the ADH1 bar was not included in the main figure in the manuscript, although the HXT bar was. Further, since HXT (and ADH1 according to the supplemental table) was chosen for expression, HXT should probably be included in the parenthetical, as production in all three cases was relatively equal. On a related note, in the paragraph referring to expressing PsCXE1, it is stated that "we engineered the 7-producing strain to express PsCXE1 downstream of a strong promoter (PYK1, GPD) and observed..." As both promoters are used in various strains as well, how did production levels compare using the different promoters?

We thank the reviewer for pointing this out. We have moved the ADH1 data previously presented in the original Supplementary Figure 2d into the main figure, and modified the indicated sentence to include a reference to HXT7 as "weak promoters (ADH1, CYC1) and late promoters (HXT7)". The synthesis of **8** in yeast strain expressing PsCXE1 downstream of PYK1 and GPD are of comparable levels as indicated in the modified Supplementary Fig. 2e.

Some of the strains seem incorrect in the supplemental table. Some examples: the table shows the same strain used for figures 2G-ii, 2H-ii, and 2I-ii, however these seem to be 3 unique strains according to the figure legend; same for the strain listed for figures 2H-v and 2I-iv.

We thank the reviewer for pointing this out and have corrected the strain information in the supplementary table or figure legend. 2h-ii and 2i-ii are EIC traces of the same **9**-producing strain expressing PsMT2, which synthesizes **8** as well, and were used as an **8**-producing strain testing the function of PsMT2 on **8**. Similarly, 2h-v and 2i-iv are both EIC traces of the **9**-producing strain expressing PsMT2 and PsMT3. We originally included detailed strain information in the Supplementary table and not the figure caption so as to avoid confusion. However, based on the reviewer's questions we realize that this may be confusing. We have modified the manuscript to indicate the complete strain information in the updated figure legend.

All final plasmids should be listed in Supplemental Table 2, not just plasmids used to create other plasmids. Further, "For plasmid-based DNA assembly, certain cassettes were PCR amplified and incorporated into pYES1L as described previously to yield..." should have a citation as this wasn't explained previously in this work. A number of plasmids appear to not have explanations for how they were made (e.g. pCS3149, pCS3173-3175).

We thank the reviewer for pointing this out and have modified the supplementary tables to include all final plasmids (the yeast expression plasmids are listed in updated Supplementary Table 2, and the remaining are listed in updated Supplementary Table 3, which was the original Supplementary Table 2). We have also included the reference (original reference 16, now reference 4 in Supplementary References) to the plasmid-based DNA assembly strategy. All the plasmids with the name of "pAG" are constructed with the gateway cloning method as described in "Construction of new designation vectors for Gateway® cloning of Plasmids and Yeast Construction" section. We have edited this section to make that point more clear.

When converting to the norlaudanoline fed strain, the enzyme norcoclaurine 6-O-methyltransferase (Ps6OMT) was added. The reviewer is curious if this has an effect on the investigated O-methylation step later in the pathway as this enzyme was shown to also create the heterodimer and catalyze the reaction to produce noscapine. It would be interesting to see what effect removing PsMT3 and lessening the heterologous burden on the cell would have on noscapine titers.

We thank the reviewer for pointing this out. We have modified the manuscript to include the data of strain (CSYN15) harboring the minimal set of enzymes for the synthesis of noscapine from norlaudanoline (AtATR1, Ps6OMT, Ps4'OMT, PsCNMT, PsBBE, TfS9OMT, CjCAS, CYP82Y1, CYP82X2, PsAT1, PsSDR1, PsTNMT, PsMT2, CYP82X1, PsCXE1) in Figure 5b.

The authors should provide the DNA sequences of the enzymes as they were optimized for expression either in the supplementary information or include their Genebank codes in the paper or methods section.

We thank the reviewer for pointing this out and have included the sequence information in the Supplementary Table 4.

In the methods section, under Culture Conditions for Metabolite Production, it appears that productions were done using galactose as a carbon source. As no enzymes were under galactose-inducible promoters, is this correct? This should be stated in the text.

We thank the reviewer for pointing this out. As indicated in the text, most of the assays were carried out using galactose as a carbon source. Our original constructs expressing CYP82Y1, CYP82X1, and CYP82X2 used GAL1 as the promoter. To compare the activities of P450s downstream of other promoters to the original GAL1 promoter, we cultured all the yeast strains under the same carbon source, galactose. In addition, the usage of galactose in alkaloid synthesis has been previously discussed in original reference 15 (now reference 47). This previous investigation indicated that when using galactose instead of dextrose as the carbon source, the production of benzylisoquinoline alkaloids from fed substrate is higher. One possible explanation provided for the increase in alkaloid production was related to certain metabolic and transcriptional responses induced by galactose in yeast. We have modified the manuscript text to clearly state the use of galactose as the carbon source.

While mentioned in the methods section, it should be mentioned in the text that some standards were unavailable and thus structurally similar standards were used to quantify titers. Further, were internal standards included in sample analysis?

We thank the reviewer for pointing this out, and have edited the manuscript text to include the suggested statements. Since we directly apply the medium or diluted medium for LC/MS analysis without multiple sample preparation steps, internal standards were not included in the sample analysis.

Figure 1. In Figure 1, it appears that Ps6OMT (in the reaction converting scoulerine to tetrahydrocolumbamine) is mislabeled as Tfs9OMT. It would also be helpful, to understand the full system from the figure, if AtATRI was included somewhere in the figure. There also appears to be a typo in the compound name of 4.

We thank the reviewer for these suggestions and have made the requested changes to Figure 1. The conversion of scoulerine to tetrahydrocolumbamine should be labeled as being catalyzed by PsS9OMT.

Figure 2. In Figure 2, include traces for standards when available.

We thank the reviewer for this suggestion and have included the trace of noscapine standard in Fig. 2i.

Figure 3. In the caption of Figure 3c, it's stated "through varying promoter and copy number"; however, only the promoters were varied in the figure. In Figure 3d, 2 should be racemic; this should be reflected in the structure and the y-axis. Also, the strains appear to be labeled incorrectly in the figure and caption (CSYN10 and 11 instead of 9 and 10 as listed in the supplemental table).

We thank the review for pointing these typos out and have made the suggested changes to the manuscript text and figures.

Reviewer #2 (Remarks to the Author):

The authors describe the construction of a strain of yeast capable of producing the alkaloid noscapine. This accomplishment is achieved through the step-wise reconstruction of an already elucidated plant pathway/gene cluster. The result is a nice accomplishment for the field of secondary metabolic production. The paper describes a unique chemistry pathway and provides proper citations for the field. This paper will be a nice contribution to the journal.

We thank the reviewer for the very supportive comments about the work.

Reviewer #3 (Remarks to the Author):

*This manuscript reports the assembly of the complete pathway for the biosynthesis of the medicinal alkaloid noscapine in yeast (*Saccharomyces cerevisiae*). This is an exciting development that follows on from the work from the Graham lab at the University of York (UK) reporting the identification and partial characterization of a ten gene metabolic cluster for the noscapine pathway in poppy. Increasingly more complex plant natural product pathways have been recently assembled in yeast, including those for opioids from poppy from the Smolke lab and also the strictosidine pathway from Madagascan periwinkle from the O'Connor lab. Despite the work from the Graham lab and others working on the noscapine pathway (e.g. the Facchini lab) the full pathway has until now not been fully characterized and questions about certain pathway steps have remained outstanding.*

The work presented here is elegant and pioneering. A series of steps was taken to engineer the ability to synthesise noscapine from canadine. Various limitations were overcome, including optimisation of the limiting step catalysed by CYP82Y1. This was achieved by optimising the N-terminal tag used for expression of CYP82Y1, the promoters and the growth temperatures. This resulted in a strain optimized for the production of compound 4, which was then used in investigations of downstream steps that provide important support for the sequence of events previously reported from in vitro studies involving CYP82X2, PsAT1 and CYP82X1. Critically, the optimized yeast strain plus these engineered and optimized subsequent steps was able to generate sufficient secoberberine intermediates for full structural characterization by NMR (which had not previously been achieved due to lack of compound availability). A key finding is that the 4'-O-methylation step appears to be catalysed by an O-methyl heterodimer. This is a very important discovery. Careful analysis is carried out in yeast to demonstrate that the two methyl transferases are indeed physically associated with each other and that neither alone is sufficient to catalyse the 4'-O-methylation step. The authors then go on to demonstrate that they are able to synthesise other protoberberine compounds by exploiting 'off-pathway' enzyme activities, and to also engineer into yeast the ability to produce noscapine from the early BIA backbone norlaudanoline.

There is no analysis of enzyme kinetics included in the work. However monitoring the effectiveness of the different pathway steps in terms of mgs product/L as the authors have done is in essence a more direct way of evaluating performance in the yeast system. The approach that they have taken is pragmatic - identifying bottlenecks and overcoming them by making and testing different design formats. The demonstration that the two methyl transferases work in yeast as a heterodimer is particularly intriguing. The applicants go to some lengths to explain in the introduction why validation of enzyme function in planta can give misleading results. However it is also the case that expression of plant genes and enzymes in heterologous hosts such as yeast may also lead to artefacts. While I do not regard it as essential for the publication of this work, investigation of whether such a heterodimer exists in poppy is an obvious priority for future work.

One minor point - Claims of priority ('This is the first...') should be avoided.

We thank the reviewer for the very supportive comments about the work. And we agree that investigation on the presence of the heterodimer in poppy is an essential part of the future work. In addition, we thank the reviewer for their point on avoiding claims of priority and that expression of plant enzymes in yeast can lead to artifacts. We have made corresponding edits to the manuscript text to remove the claim of priority and clarify the point that expression of plant genes in heterologous hosts and *in vitro* can lead to artifacts (see Discussion session, 6th paragraph).

REVIEWERS' COMMENTS:

Reviewer #1 (Remarks to the Author):

The authors have thoroughly addressed all the comments raised.

I think the revised manuscript is not only technically stronger, but the importance of the manuscript now really comes across. I believe the manuscript is ready for publication.